# Integrated photonics enabling ultra-wideband fibre–wireless communication

Yunhao Zhang[1,2,3,9], Haowen Shu[2,4,9 ✉], Yijun Guo[2,9], Peiqi Zhou[5,9], Luyu Wang[6,9], Jianyang Cai[2], Liyuan Yao[2], Linshan Yang[2], Linze Li[6], Tianyu Long[6], Zhouze Zhang[6], Changhao Han[2], Kaihang Lu[7], Yu Sun[8], Zhaopeng Xu[1], Jun Qin[8], Yeyu Tong[7], Zhixue He[1], Xi Xiao[5], Lei Wang[1], Baile Chen[6 ✉], Shaohua Yu[1 ✉] & Xingjun Wang[1,2,4 ✉]

Telecommunication systems are evolving towards ultrawide bandwidth and low latency, supporting wired and wireless links and their non-blocking interconnection[1]. However, a long-standing bandwidth mismatch between fibre communication and its wireless counterpart arises from fundamental disparities in signal architectures and hardware constraints[2,3], which prevent high-speed and compatible transmission across the two domains. This challenge further complicates unified system design and hinders the realization of high-throughput-density, congestion-free fibre–wireless links under wideband-access scenarios[4]. Here we present an ultra-wideband (UWB) integrated photonics scheme that facilitates fibre–wireless communication over a shared-bandwidth infrastructure. Built on electro–optic (EO) and optic–electro (OE) conversions featuring 3-dB operational bandwidths exceeding 250 GHz and cross-architecture adaptability, our system demonstrates unprecedented data transmission capabilities in both wired and wireless links. Using the same set of devices and powered by the proposed complex bidirectional gated recurrent unit (complex-biGRU) algorithm, ultrahigh single-lane data rates of 512 Gbps for short-reach fibre and, for the first time to the authors' knowledge, 400-Gbps high-speed wireless transmission have been achieved. Furthermore, high-density access is enabled by an all-optically assisted ultra-broadband wireless scheme. Real-time multichannel 8K video transmission is successfully demonstrated across 86 channels, seamlessly using a spectral range from 138 to 223 GHz. These findings in unified telecommunication development show the potential for the development of high-speed, densified and low-latency communication networks.

Optical telecommunication has profoundly accelerated the development of massive data interconnection[3,5] and high-performance computing[6,7]. Despite great success in fibre communications, the future Internet of Everything demands high-throughput, wide-area coverage and low-latency signal delivery across all scenarios, as shown in Fig. 1, including high-speed fibre interconnections, multi-access wireless networks and hybrid fibre–wireless systems. To meet these demands, both fibre and wireless links require continuous improvement in single-lane speed and available bandwidth to enhance transmission capacity. Also, access architectures should migrate to the terahertz (THz) band and be deeply integrated with fibre technology, forming THz fibre–wireless networks to enable massive connectivity and low-cost signal processing[8]. Moreover, to cover areas beyond the reach of fibre infrastructure, such as remote habitats or harsh environments, wireless transmission links serving as hybrid relay nodes often require seamless fibre–wireless–fibre

signal conversion, referred to as transparent relaying, to ensure low-latency and high-fidelity transmission across the interface[1].

However, delivering comparable high-speed and low-latency data transmission ubiquitously across all scenarios remains a substantial challenge. One notable obstacle lies in the limited device operational bandwidth. Although baseband transceivers for fibre communications have already reached 100 GHz and beyond[9–11], supporting transmission rates of 400 Gbps per lane[12–14], such operation is already close to the bandwidth limit of present transceiver technology. In a unified fibre–wireless communication system, directly forwarding these wideband baseband signals to the wireless domain requires upconversion to the THz range (above 0.1 THz), which imposes greater challenges on the device broadband performance at the transmitter and receiver sides. Specifically, this requires a flat electro–optic–electro frequency response spanning several hundred GHz in the THz range, along with high saturation power

[1]Peng Cheng Laboratory, Shenzhen, China. [2]State Key Laboratory of Photonics and Communications, School of Electronics, Peking University, Beijing, China. [3]School of Electronic and Computer Engineering, Peking University Shenzhen Graduate School, Shenzhen, China. [4]Frontiers Science Center for Nano-optoelectronics, Peking University, Beijing, China. [5]National Information Optoelectronics Innovation Center, China Information and Communication Technologies Group Corporation, Wuhan, China. [6]School of Information Science and Technology, ShanghaiTech University, Shanghai, China. [7]Microelectronics Thrust, The Hong Kong University of Science and Technology (Guangzhou), Guangzhou, China. [8]Key Laboratory of Information and Communication Systems, Beijing Information Science and Technology University, Beijing, China. [9]These authors contributed equally: Yunhao Zhang, Haowen Shu, Yijun Guo, Peiqi Zhou, Luyu Wang. ✉e-mail: haowenshu@pku.edu.cn; chenbl@shanghaitech.edu.cn; yush@cae.cn; xjwang@pku.edu.cn

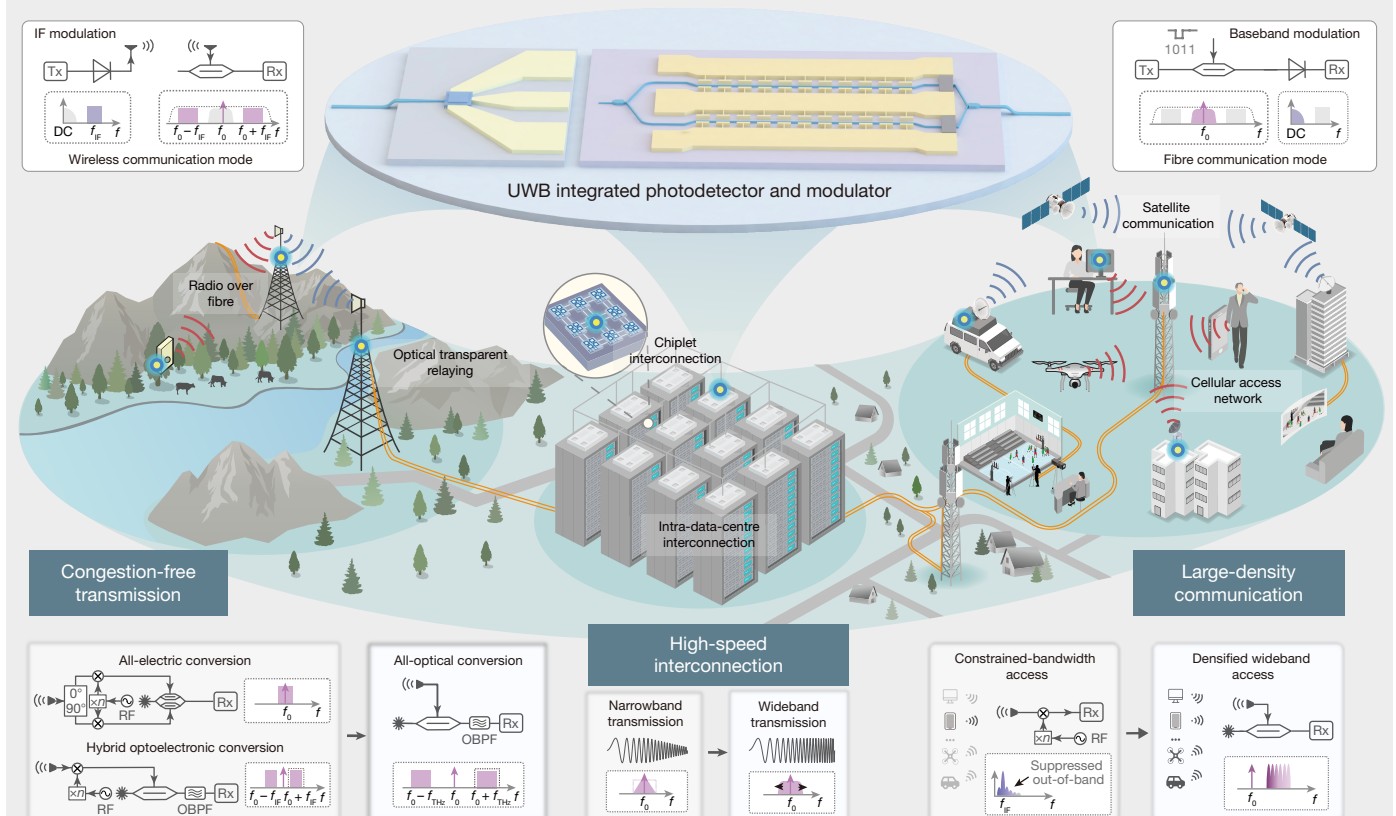

**Fig. 1 | Integrated UWB all-optical telecommunication system.** Conceptual drawings for all-optical ultra-broadband telecommunication connections (fibre–wireless congestion-free transmission, high-speed fibre interconnection and large-density fibre–wireless access) powered by the integrated photonics system. With the UWB integrated devices and seamless integration of the fibre–wireless system, high-throughput and low-latency all-optical telecommunication can be expected. OBPF, optical band-pass filter.

in photodetection and high modulation efficiency, to achieve higher speed while preserving signal fidelity, which still remains difficult to meet. Although plasmonic modulators have been proposed to achieve near-THz bandwidth and support signal modulation deep into the THz range, they suffer from low modulation efficiency and high optical loss, which may limit the higher achievable lane speed[15]. On the other hand, uni-travelling carrier photodiodes (UTC-PDs) have become the dominant technique for chip-scale optical-THz signal generation[16–18]. Yet, relatively low OE bandwidth and limited sub-mW-level saturation power restrict the signal-to-noise ratio in THz applications.

Another challenge concerns system architecture. Because electro–optic–electro signals conversions are usually processed at baseband in fibre communication and at intermediate frequency (IF) in wireless transmission, respectively, fibre–wireless systems usually require cross-band frequency mixing. To achieve this, both all-electric and hybrid optoelectronic approaches perform frequency mixing using frequency-multiplied electrical local oscillators (LOs)[19–24]. However, these approaches introduce electrical bandwidth constrains, further frequency-multiplication-induced noise accumulation and hardware complexity, which, in turn, limit system capacity and increase implementation cost. All-photonic-assisted wireless schemes have enabled direct wireless-to-optical signal conversion and subsequent processing in the optical domain, offering substantial hardware savings and excellent frequency conversion consistency[2,25,26]. Full link functionalities and broadband spectrum adaptability have also been demonstrated on an integrated photonics platform for all-optical wireless communications[27]. Nevertheless, the demonstrated single-lane data rates so far remain limited to less than 80 Gbps (ref. 25). Although various multiplexing techniques can further enhance overall capacity[26], such aggregation introduces considerable complexity in signal encoding

and decoding, thereby hindering the realization of low-latency transparent relaying. Furthermore, signal distortions caused by both linear and nonlinear impairments become increasingly pronounced as the signal bandwidth increases towards the 100-GHz range. This makes traditional linear digital signal processing (DSP) algorithms largely ineffective for broadband fibre–wireless convergence, especially as future infrastructure demands lane rates exceeding 400 Gbps.

Here we present a unified UWB fibre–wireless communication solution based on integrated photonics. Using a pair of photonic EO and OE converters with state-of-the-art bandwidths exceeding 250 GHz, consisting of a thin-film lithium niobate (TFLN) modulator and a modified UTC-PD, we realize a fibre–wireless bandwidth-shared transmission scheme with more than 100 GHz channel bandwidth available in both the fibre and wireless links. Facilitated by efficient signal modulation, high-power photodetection and a unified complex-biGRU algorithm, our system achieves high-quality data transmission in both fibre and wireless links. The single-lane data rate is boosted to, to the best of our knowledge, the highest levels in both scenarios, with 512 Gbps achieved over the fibre link and 400 Gbps over the wireless link at the THz band. More practically, an 86-channel 8K real-time video transmission has been realized with 1-GHz channel bandwidth across 138–223 GHz, which is one order of magnitude larger than the standard 5G protocol. Our work paves the way towards the full integration of ultra-broadband all-optical communication systems and will be a promising route for next-generation telecommunications.

## Ultra-broadband on-chip EO and OE conversion

The ultra-broadband ability of our demonstration is ensured by a TFLN modulator and an indium phosphide (InP)-based UTC-PD, both

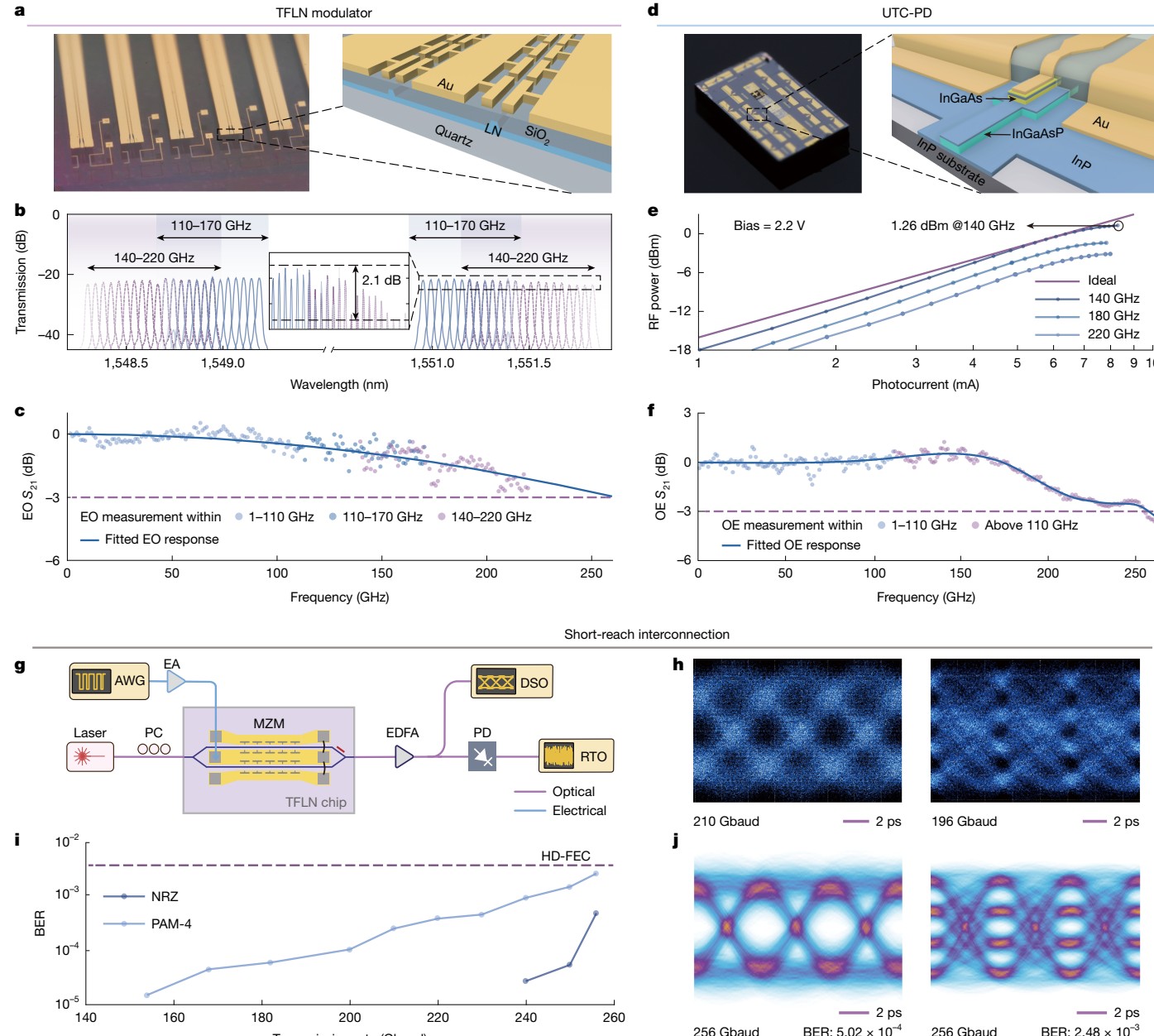

**Fig. 2 | Characterizations of fundamental UWB building blocks and short-reach interconnection. a**, Optical image and 3D cross-section view of the TFLN modulator. **b**, Optical spectrum of the modulated sidebands (carrier omitted). The bandwidth measurement uncertainty from 110 to 220 GHz is ±0.4 dB owing to the power uncertainty of the OSA. **c**, Normalized EO response from 1 to 220 GHz. The solid line is the extrapolated response by fitting experimental data. **d**–**f**, Optical image and 3D schematic diagram (**d**), RF output power (**e**) and OE bandwidth (**f**) of the modified UTC-PD. **g**, Schematic of the IMDD data transmission set-up. EA, electrical amplifier; PC, polarization controller. **h**, Eye diagrams of 210-Gbaud NRZ (left) and 196-Gbaud PAM-4 (right) captured by the DSO. **i**, BER for NRZ and PAM-4 transmission at different symbol rates using the complex-biGRU algorithm. All results are considered within the given HD-FEC threshold. **j**, Reconstructed eye diagrams and BERs of 256-Gbaud NRZ and PAM-4 signals.

of dedicated design and fabricated for large bandwidth and efficient conversion. The device photograph and cross-section schematic diagram of the integrated EO modulator are shown in Fig. 2a. The chip is fabricated on a 360-nm X-cut lithium niobate (LN) wafer, with a 500-µm-thick quartz substrate used to reduce microwave loss (Methods). To compensate for the slow wave effect induced by quartz substrate, a periodic capacitively loaded travelling-wave electrode (that is, slow-wave electrode) structure is applied[28], with excellent velocity match and impedance match (Supplementary Note 1). The input and output pads are also appropriately designed to have a 50-Ω characteristic impedance as well as 50-Ω on-chip terminators at the end of the capacitively loaded travelling-wave electrodes.

We then show the first >200 GHz 3-dB EO response characterization of a TFLN modulator. Including the lensed fibre-to-chip coupling loss (about 3.8 dB per facet), the total optical loss comes to about 8.2 dB, indicating an on-chip insertion loss of 0.6 ± 0.3 dB. The EO response is measured using a lightwave component analyser (LCA) under 110 GHz and optical spectrum analysis for higher frequencies (Methods). Figure 2b shows the sidebands of the modulated signal in the range 140–220 GHz. The inset depicts a flat optical spectrum over the test range with a ripple of 2.1 dB. The whole EO response shows an ultrahigh experimental bandwidth of >220 GHz (Fig. 2c) and an extrapolated result of about 260 GHz (with respect to 1 GHz). To the best of our knowledge, this is the highest experimentally measured EO bandwidth

reported for TFLN modulators, excluding previously extrapolated or estimated results (Extended Data Table 1). Benefiting from the excellent EO bandwidth, radio frequency (RF) half-wave voltage $V_{\pi,RF}$ of 6.2 V at 200 GHz can be calculated (Methods). To ensure both fibre and wireless communications, the device is designed to achieve better balance between EO bandwidth and modulation efficiency while maintaining lower insertion loss. Compared with other EO bandwidth enhancement techniques[15,29–31], our design also obtains a flat EO response with the maximum deviation of 0.5 dB above the 0 dB level. Such property offers substantial improvements towards faithful high-speed wireless communication requiring wide spectral occupancy and uniform signal integrity in multichannel real-time video transmission.

The optical image and corresponding schematic diagram of the UTC-PD are presented in Fig. 2d. The epitaxial structure of the device is grown on a semi-insulating InP substrate[17], incorporating a modified uni-travelling carrier structure with a partially depleted InGaAs absorber and a lightly n-doped InP collector. To enhance the high-speed performance of the device, a 30-nm heavily n-doped ($2 \times 10^{17}$ cm$^{-3}$) InP cliff layer is introduced in the drift layer to modulate the internal electric field. This design enables broader device bandwidth while maintaining a high saturation output power. The frequency response and saturation power are characterized by an optical heterodyne set-up and power meters (Methods). As shown in Fig. 2e, the device with a compact $2 \times 10$-µm$^2$ active region delivers up to 1.26 dBm at 140 GHz, with a 1-dB compression point at 8 mA. In terms of OE bandwidth, the device exhibits a relatively flat frequency response from DC to 170 GHz and achieves a 3-dB bandwidth exceeding 250 GHz (Fig. 2f and Extended Data Table 2). Also, the measured dark current of the device is less than 100 pA under −5-V bias, thereby suppressing the system noise floor and making the device well suited for high-speed communication systems.

The proposed UWB devices are first used in fibre interconnection scenarios. Several intensity modulation direct detection (IMDD) tests under non-return-to-zero (NRZ) and pulse-amplitude four-level modulation (PAM-4) formats are conducted. The experimental set-ups are shown in Fig. 2g. At the Tx end, a periodic pseudorandom bit sequence signal is generated by an arbitrary waveform generator (AWG) and then encoded on a 1,550-nm optical carrier. A digital sampling oscilloscope (DSO) equipped with a 120-GHz calibrated optical sampling module is used to directly capture the modulated signal (Methods), as shown in Fig. 2h. Clear eye opening has been observed at different symbol rates up to 210 Gbaud for NRZ and 196 Gbaud for PAM-4 (Supplementary Note 3), representing the highest speed without bandwidth compensation DSP so far. The modulated signals are also recorded by a real-time oscilloscope (RTO) and demodulated using the offline DSP method. By applying the complex-biGRU algorithm (details in the next section and Methods), 256-Gbaud NRZ and PAM-4 transmission have been realized with bit error ratios (BERs) of $5.02 \times 10^{-4}$ and $2.48 \times 10^{-3}$, respectively (Fig. 2i,j). The realized BER is much lower than the 7% hard-decision forward error correction (HD-FEC) threshold of $3.8 \times 10^{-3}$ and primarily limited by the sampling rate of the measurement instruments. To our knowledge, this is the highest single-lane symbol rate and PAM-4 data rate. Using a much lower order modulation format, a state-of-the-art net bit rate of 479 Gbps can be calculated after forward error correction (FEC) overhead subtraction (Extended Data Table 3).

## THz all-optical wireless transparent relaying

Apart from baseband fibre transmission, UWB devices have sufficient IF bandwidth with efficient EO/OE response, extending up to the THz band, which advances wireless communication speed for congestion-free fibre relaying. Here high-speed wireless optical transparent relaying is performed with a wireless carrier around 180 GHz. Figure 3a illustrates the overall architecture of the transceiving system (Methods). For THz signal generation, a commercial silicon coherent transmitter is used to generate the optical baseband and a tunable external cavity

laser (ECL) with a frequency offset of 180 GHz is used as the Tx LO. Two horn antennas are placed at a distance of 20 cm for THz emitting and receiving, which could be further enhanced by incorporating a lensed antenna to improve beam focusing and coupling efficiency. Owing to the high saturation power of the modified UTC-PD and low $V_{\pi,RF}$ of the TFLN modulator, only one low-noise amplifier (LNA) is used at the Rx side to amplify the attenuated THz signal, which subsequently drives the TFLN modulator, allowing for future direct signal demodulation using mature optical DSP.

To accurately characterize the intrinsic performance, the single-channel rate, defined as the aggregate wireless air interface rate divided by the product of the numbers of active space channels, polarization states and carrier wavelengths, serves as the primary comparative metric. Several modulation formats including quadrature phase-shift keying (QPSK), 16-QAM and 32-QAM (QAM denoting quadrature amplitude modulation) at different speeds have been used in our experiments. Figure 3b depicts the calculated BERs of all test conditions. After signal recovering on the basis of baseline DSP, all BER results remain below the 20% soft-decision forward error correction (SD-FEC) threshold ($4 \times 10^{-2}$), with the highest transmission rates up to 180, 240 and 180 Gbps for QPSK, 16-QAM and 32-QAM, respectively (Fig. 3c). We also verify the performance of the system at a 4-m wireless distance using lens antennas. All BER results meet the SD-FEC threshold requirement, with the highest transmission rates up to 192 Gbps for QPSK and 304 Gbps for 16-QAM (Methods and Supplementary Note 4). Despite using only baseline algorithms, our system still reaches higher single-channel symbol rate and data rate compared with previous THz communications approaches with advanced DSP algorithms[21].

To further alleviate the influence of increasing linear and nonlinear signal distortion in the wide signal band, the complex-biGRU algorithm has also been proposed to push the performance of the system to its limit (Fig. 3d). Our scheme expands the traditional gated recurrent unit (GRU) network into the complex domain to adapt for coherent optical communications. To avoid signal distribution distortion brought by the neural network (NN)[32,33], a multilevel activation function is implemented and the 'jail window' effect is notably relieved (Methods and Supplementary Note 5). Applying this algorithmic equalization, a series of transmission tests is carried out and the resulting BER performances are shown in Fig. 3e. Ultrahigh-speed THz communication up to 400 Gbps within SD-FEC is achieved for both 16-QAM and 32-QAM. For QPSK, 100-Gbaud transmission has been realized with the BER less than $1 \times 10^{-5}$, reaching the BER floor of our experiment. Distinguishable constellation diagrams of the highest speed for different types of signal are observed in Fig. 3f. We also achieve 400-Gbps transmission for both QPSK and 16-QAM at a 4-m wireless distance using lens antennas (Methods and Supplementary Note 4). To the best of our knowledge, these results represent the highest single-channel symbol rate and data rate for all THz communications (Extended Data Table 4).

Powered by the complex-biGRU algorithm, the proposed UWB integrated photonics system first breaks the 100-Gbaud and 400-Gbps barriers, exceeding the previous demonstrations by factors of 1.33 and 1.53, respectively. Furthermore, our system also reports 213% enhancement in carrier utilization efficiency, surpassing the conventional limit of 1 for the first time and reaching 2.22, thereby setting a new benchmark in the high-speed THz communication field. It is also noteworthy that both the UWB integrated photonics devices and the complex-biGRU algorithm are compatible for both wired and wireless communications and can therefore serves as universal functional units to support dual-mode transmission.

## All-optical multi-user wireless access

Beyond substantially enhancing single-carrier transmission data rates, the UWB integrated photonics system also promises applicability in multi-user-access scenarios. In our proof-of-concept demonstrations,

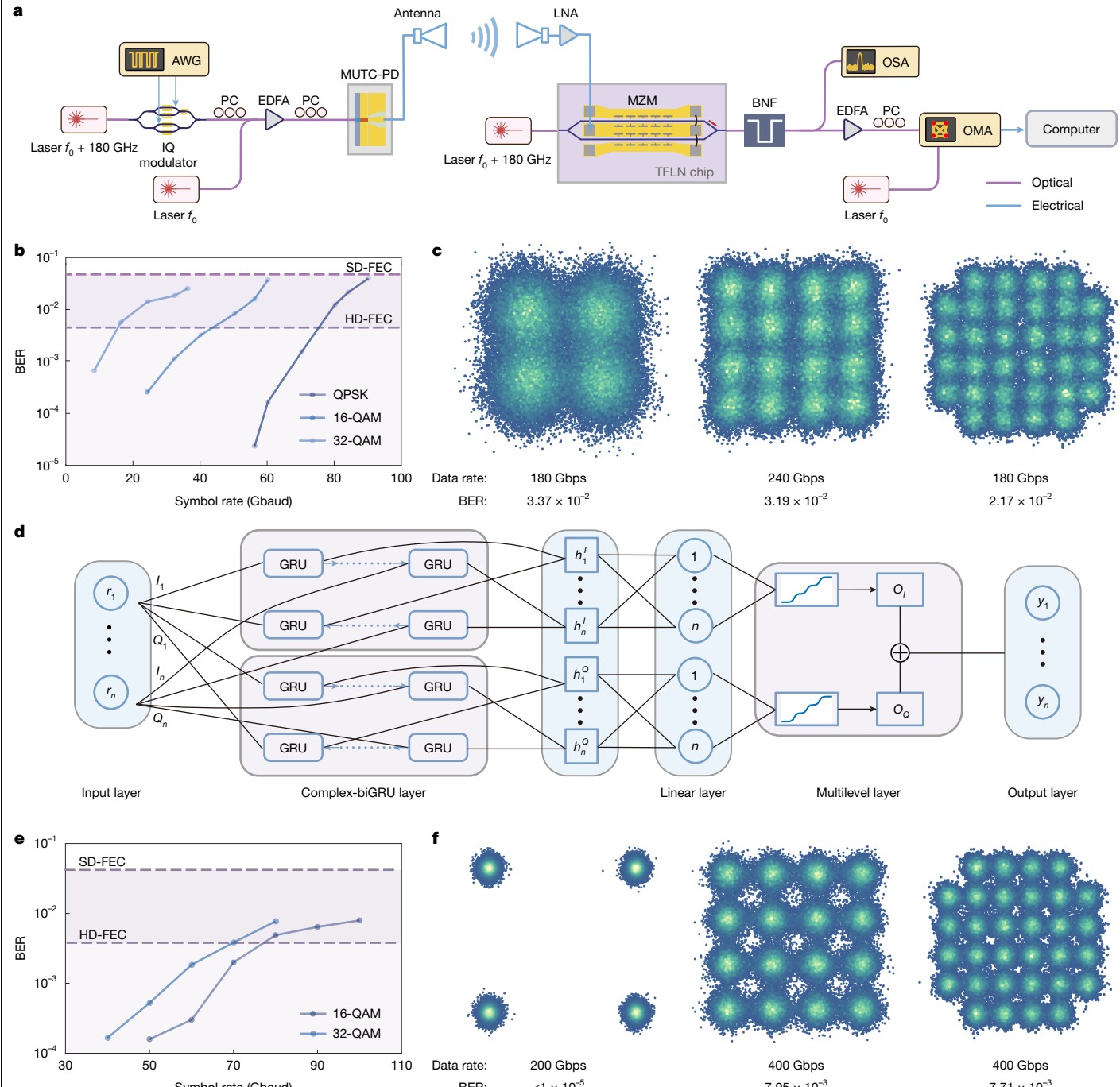

**Fig. 3 | High-speed THz wireless transmission results. a**, Schematic of the UWB all-optical THz wireless transmission set-up. The blue and purple lines indicate the electrical and optical links, respectively. The carrier frequency used in the experiment is 180 GHz. All of the data are collected and processed offline. BNF, band notch filter; MUTC-PD, modified UTC-PD; OMA, optical modulation analyser; PC, polarization controller. **b,c**, BER results (**b**) and constellation diagrams (**c**) at different symbol rates (10–90 Gbaud) with the baseline DSP method. All results are under the given SD-FEC threshold.

Constellation diagrams are only shown at the highest symbol rates for different modulation formats. Other results can be found in Supplementary Note 4. **d**, Architecture of the proposed complex-biGRU algorithm. For each equalization, the network parameters are initially optimized using one set of data, followed by inference generation on another dataset. **e,f**, BER results (**e**) and constellation diagrams (**f**) at different symbol rates (40–100 Gbaud) with the complex-biGRU algorithm. All results are far below the SD-FEC threshold, showing a marked enhancement brought by the complex-biGRU algorithm.

86 channels of real-time streaming 8K videos transmission are realized. The schematic diagram of the proposed system is sketched in Fig. 4a (Methods). The uplink servers connect to the Tx switch through Ethernet ports with the standard TCP/IP protocol. After adding FEC overhead, the video streams are loaded onto off-the-shelf SFP+ modules and aggregated by a passive optical multiplexer (optical coupler in this case) as the composite optical signal. This signal then undergoes

the same optical–THz–optical conversion as we demonstrated above. For the downlink route, the composite optical signal is separated into baseband signal using a passive optical demultiplexer (optical coupler and filters in this case) and distributed by the Rx switch to different clients according to the preset configuration.

To verify the multi-user-access capability, the Tx LO frequency offset is adjusted from 138 to 223 GHz with a 1-GHz step. As a result, consecutive

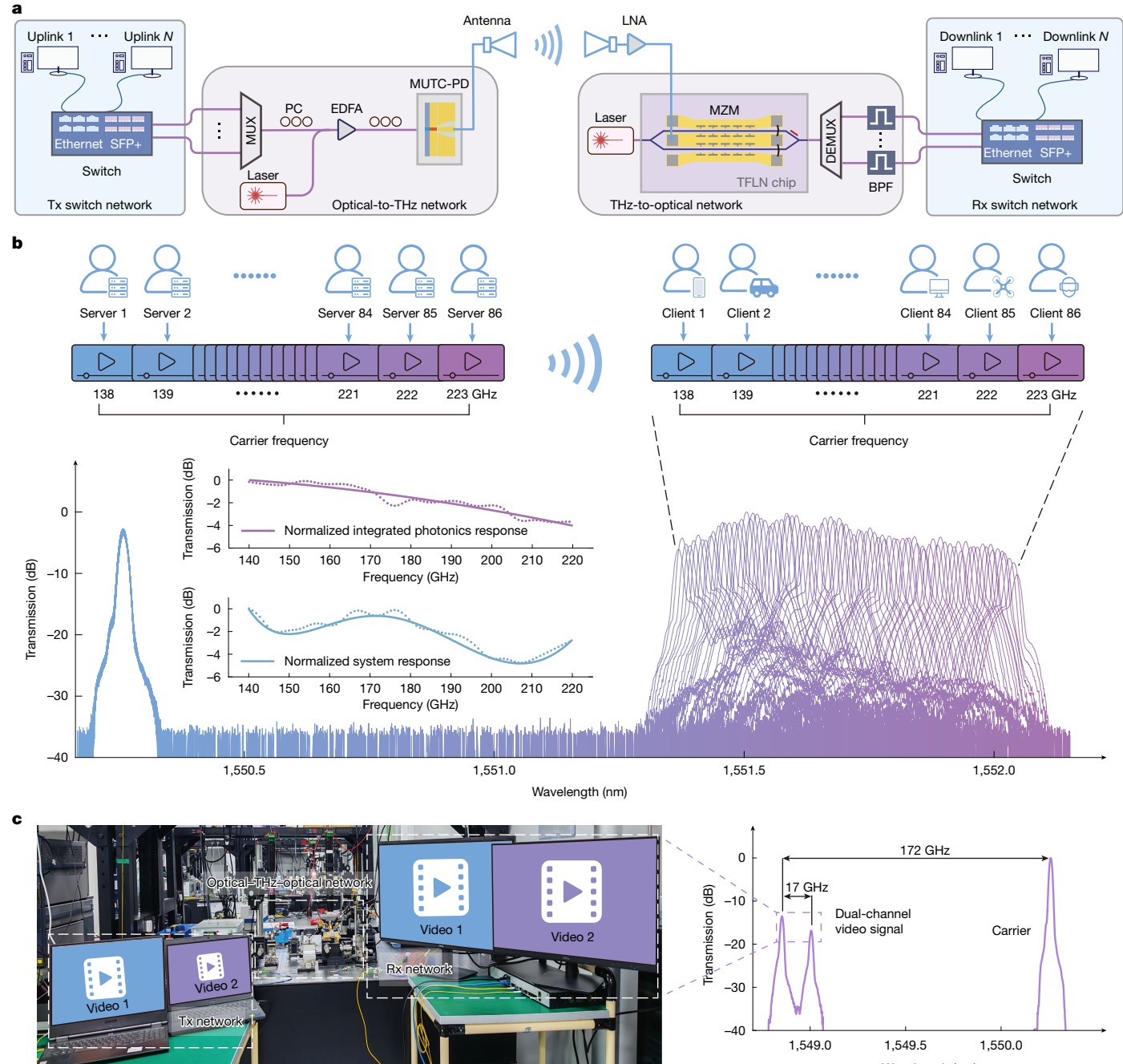

**Fig. 4 | Multichannel real-time videos transmission results. a**, Schematic of the multi-user-access system. The entire system can be divided into four subsystems: Tx switch network, optical-to-THz conversion network, THz-to-optical conversion network and Rx switch network. BPF, band-pass filter; DEMUX, demultiplexer; MUX, multiplexer; MUTC-PD, modified UTC-PD; PC, polarization controller. **b**, Proof-of-concept demonstration of multichannel real-time videos transmission. Each video represents an individual server/client allocated to the corresponding channel from 138 to 223 GHz with 1-GHz bandwidth. The aggregated optical spectrum shows the received signal spectrum for each channel. Insets, directly measured integrated photonics system response (top) and normalized system response extracted from the whole links (bottom). **c**, Proof-of-concept demonstration of parallel real-time videos transmission. Left, photograph of the experimental set-up. Right, optical spectrum of the received videos signals. The imbalance of the received power between two channels mainly results from the power disparities of the original Tx signals.

and clear live video is displayed on the receiver screen for all channels. The optical spectrum and screenshot corresponding to each channel are shown in Fig. 4b. The lower inset illustrates the normalized system response extracted from the envelope of different channels after de-embedding other components in the transmission link, showing a low signal degradation (<5 dB) brought by the UWB integrated photonics system owing to relatively high optic–electro–optic conversion efficiency. The consistent response measured for the proposed system

(upper inset) confirms the system's advantages of broad bandwidth and flat frequency response. Our approach allocates 1 GHz of bandwidth for each user, offers 10–20 times improvement over standard 5G communications[34] and satisfies most 6G consumer application demands. We also conduct real-time video transmission at a 4-m wireless distance and clear live video can also be seen on the receiver screen (Supplementary Video 1).

The parallel connection ability is further assessed, with two individual servers used as the Tx servers. Limited by the resolution and accuracy

of the fibre Bragg grating (FBG) filters and SFP+ modules used in the experiment, two video stream signals are modulated onto 155-GHz channel and 172-GHz channels simultaneously. Passing through the same all-optical access network, the received videos, identical to the transmitter, are shown on the laptops (that is, Rx clients) respectively (Fig. 4c). Compared with the previous digital coherent optical approach[4], our scheme uses IMDD transceivers and passive optical network, providing several optimizations in the system's power consumption, expenditure, complexity and maintainability.

## Discussion

In this work, we propose and demonstrate a prototype system for UWB all-optical telecommunications, with ultrahigh lane speeds demonstrated in both short-reach fibre and all-optical wireless links. Such a scheme will provide an effective solution to the applications that simultaneously require ultrahigh-speed data transmission for both fibre and wireless scenarios, such as 6G base stations[35] and wireless data centres[36]. The performance of these systems can be improved through further optimizations. At present, the transmission rate is mainly limited by the analogue bandwidth of the probes, amplifiers, antennas and other test equipment. We are fully convinced that our system can support a higher single-channel rate and single-lane rate with development in these areas. For THz communications, the proposed system can be further scaled up to >10 Tbps by using multiplexing techniques including polarization multiplexing for both optics and antennas[37], frequency division multiplexing at higher carrier frequencies[38], spatial multiplexing with multiple-input–multiple-output structures[21], as well as advanced signal coding schemes such as probability shaping[21] and orthogonal frequency division multiplexing[25]. By exploiting high-gain Cassegrain antennas (>50-dBi gain)[20,25], polytetrafluoroethylene lens (>50-dBi equivalent gain)[39,40] and cascade THz amplifiers at both the Tx and the Rx sides[2], the coverage distance can be extended to 1 km or more.

The proposed integrated photonics telecommunication system not only has the advantage of large capacity but also shows exceptional performance in other essential features. (1) Power-efficient: low insertion loss, low $V_{\pi,RF}$ and high saturation power alleviate the demand for high-power amplifications in the communication links. (2) Low cost: all-optical link eliminates further electrical components in the transmission, saving the budget for purchasing extra RF wire and mixer. No extra customized manufacturing processes are needed for fabricating the two crucial devices. (3) Massive scalability: the design parameters of two devices comply with commercial wafer-scale fabrication requirements and are capable of volume production. All-optical architecture enables seamless integration with existing optical networks. (4) High flexibility: THz transmission at different carrier frequencies can be easily achieved by simply adjusting the Tx and Rx LOs without any mixing bandwidth limitation. The complex-biGRU algorithm offers exceptional adaptive capabilities, making the system resilient to diverse complex channel conditions.

We also anticipate more cooperations with integrated lasers[41], on-chip antennas[42], tunable microwave photonic filters[43,44], waveguide amplifiers[45] and heterogeneously integrated photodiodes (PDs)[46] based on the TFLN platform, eliminate discrete instruments and culminate in monolithically fully integrated optoelectronic systems. Beyond extraordinary communication performance, the versatility of our strategy provides potential advancement to diverse domains. For instance, chip-scale microwave frequency measurements and manipulations can be broadened into the THz regime based on UWB microwave photonic circuits[47]. Integrated radio detection and ranging (radar) operating in the THz band with high resolution and real-time sensing facilitates 6G applications such as indoor positioning and vital-sign monitoring[48]. For THz spectroscopy and imaging, our scheme provides compact and affordable THz generation, modulation and detection, which are critical for industrial fault analysis and biological diagnosis[49].

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

## Methods

### Design and fabrication of the integrated modulator and modified UTC-PD

The TFLN modulator is fabricated on a 360-nm-thick X-cut single-crystalline LN thin film bonded on a 2.5-µm silicon dioxide layer sitting on a quartz substrate (NanoLN). First, the LN waveguides and multi-mode interference structures are patterned by positive resist by means of electron-beam lithography and etched by fluorine-based reactive ion etching. After the first lithography and etching, the LN waveguides with a sidewall angle of 72°, slab thickness of 180 nm and rib height of 180 nm are obtained. The slab is then patterned and etched to form the edge couplers. After patterning the LN waveguides and edge couplers, a plasma-enhanced chemical vapour deposition $SiO_2$ film of thickness 1.2 µm is deposited as the cladding layer. Subsequently, a 180-nm-thick NiCr layer is deposited as the terminal resistor and on-chip terminator. Next a lift-off process produces the 1-µm-thick gold transmission lines and DC electrodes. Finally, the device end face is diced, lapped and polished to achieve improved end-face coupling efficiency. The fabrication also complies with standard lithography and etching techniques available in a typical foundry, ensuring process reproducibility and potential for scalability.

The epitaxial structure of the modified UTC-PD is designed for high-speed and high-power operation. The 180-nm InGaAs absorption layer includes an 80-nm undepleted region with a step-graded profile ($5 \times 10^{17}$, $1 \times 10^{18}$ and $2 \times 10^{18}$ cm$^{-3}$) to form a quasi-electric field that assists electron transport and the remaining 100 nm is depleted to establish a high electric field across the InGaAs/InP heterojunction. Beneath the absorber, a 220-nm slightly n-doped ($1 \times 10^{16}$ cm$^{-3}$) InP drift layer is incorporated to support high-speed carrier transit. A 30-nm heavily doped ($2 \times 10^{17}$ cm$^{-3}$) InP cliff layer is inserted between the absorber and the collector to sustain an electric field of 20–50 kV cm$^{-3}$ in the drift layer to support electron velocity overshoot, thereby accelerating electron transmission and extending device bandwidth. The implementation of the cliff layer also greatly reduces the conduction band barrier across the heterojunction in the depletion region, thus suppressing carrier pile-up under high optical input and improving high-power performance. To further suppress parasitic capacitance and enhance RC-limited bandwidth, a 4-µm-thick benzocyclobutene dielectric layer is introduced beneath the coplanar waveguide electrodes, improving electrical isolation and device robustness. Moreover, the InP drift layer is extended outward to serve as the upper cladding of the InGaAsP waveguide, ensuring uniform optical absorption and mitigating localized saturation under strong light injection. Also, the thickness and refractive index of the 400-nm InGaAsP waveguide layer is carefully engineered to enable efficient and smooth evanescent light coupling into the absorber. This results in a more evenly distributed light absorption and carrier generation profile across the absorber, which mitigates localized saturation and enables higher output power under strong light injection.

The fabrication of the modified UTC-PD begins with the epitaxial growth on a 2-inch semi-insulating InP substrate using metal–organic chemical vapour deposition. The epilayers are similar to the structure in ref. 17. Ti/Pt/Au metal layers are then deposited to form the p-type ohmic contacts. The triple-mesa structure of the device is defined through a combination of inductively coupled plasma dry etching and selective wet etching. The etch process is precisely controlled to ensure accurate patterning, enabling close agreement between optical simulations and the fabricated device. Subsequently, Ge/Au/Ni/Au layers are deposited to form the n-type ohmic contacts, followed by rapid thermal annealing at 360 °C to enhance contact performance. Finally, a 3-µm-thick layer of benzocyclobutene is applied to provide passivation and mechanical support, following which Ti/Au coplanar waveguides are deposited to connect the electrodes to the PD contacts. The process flow uses a standard III–V semiconductor process and is compatible with wafer-scale manufacturing.

### Characterization of the EO/OE response of the TFLN modulator and modified UTC-PD

We use a function waveform generator (RIGOL DG922 Pro) to generate a 100-kHz triangular voltage sweep and measure the $V_{\pi,LF}$ as 5.1 V by a digital oscilloscope (RIGOL DHO1202U). The $V_{\pi,RF}$ can then be calculated using:

$$V_{\pi,RF} = V_{\pi,LF} \times 10^{-\frac{EO\,S_{21}}{20}} \tag{1}$$

As for the EO bandwidth characterization, measurements are conducted in three steps. First, a vector network analyser (Keysight N5222B) equipped with a millimetre test set (Keysight N5292A) and a LCA (Keysight N4372E) is used to generate and analyse frequency signals below 110 GHz. For higher frequencies, THz signals >110 GHz are generated by upconversion. The 250-kHz to 20-GHz signals (Keysight E8257D) are upconverted by frequency multipliers into the 110–170-GHz band (NZJ SGFE06) and the 140–220-GHz band (NZJ SGFE05) and then delivered to the TFLN modulator by means of matched ground–signal–ground (GSG) probes. EO response is measured by tracking the power ratio between the sideband and the carrier at each frequency using an optical spectrum analyser (OSA; Yokogawa AQ6370D). The normalized sideband power $P_n$ is defined as:

$$P_n = \frac{P_{carrier}}{P_{sideband}} \tag{2}$$

The $V_{\pi,RF}$ at each frequency can be calculated directly using equation (3) (ref. 50):

$$V_{\pi,RF} = \frac{1}{4}\pi V_p \sqrt{P_n} \tag{3}$$

in which $V_p$ denotes the maximum voltage of the input THz signal. The EO response can be derived from equations (1)–(3) as:

$$EO\,S_{21} = -20 \times \log\left(\frac{\pi}{4V_{\pi,LF}}\right) - 20 \times \log(V_p) - P_c + P_s \tag{4}$$

The first term of equation (4) represents the constant term related to $V_{\pi,LF}$. The second term is the drive power delivered to the modulator and can be calculated by subtracting the probe loss from the output power of the frequency multipliers. $P_c$ and $P_s$ are the carrier and sideband power, respectively, measured from the OSA corresponding to each test point. After de-embedding the difference of the driving power at each test frequency, the EO response is proportional only to the carrier and sideband amplitudes. Because the probes and frequency multipliers have been precisely measured and calibrated, the primary source of measurement uncertainty lies in the power uncertainty of the OSA. According to the operation manual, the power measurement uncertainty of the OSA is ±0.4 dBm, resulting in a bandwidth measurement uncertainty of ±0.4 dB from 110 to 220 GHz. The calculated results for the 110–170-GHz and 140–220-GHz bands are first normalized by their overlapping-region average and then referenced to the 110-GHz data from the LCA.

The frequency response and saturation behaviour of the modified UTC-PD are characterized using an optical heterodyne set-up. Two tunable lasers with wavelengths $\lambda_1$ and $\lambda_2$ are combined to generate a tunable beat frequency $f_{beat}$ by fixing one wavelength and tuning the other. Polarization controllers ensure optimal alignment, enabling nearly 100% modulation depth. The combined optical signal is amplified by an erbium-doped fibre amplifier (EDFA) and then coupled into the chip. A DC bias is applied to the device using a source meter (Keysight B2901A).

$$f_{beat} = f_1 - f_2 = \frac{c}{\lambda_1} - \frac{c}{\lambda_2} \tag{5}$$

For RF measurements, the RF signal generated by the UTC-PD is directly measured by a power meter through a GSG probe. Under 100% modulation depth, the ideal RF output power follows equation (6) with $R_L = 50\,\Omega$, corresponding to the purple reference line in Fig. 2e. Different measurement set-ups are used for different frequency ranges to ensure accurate power measurement. From DC to 110 GHz, output power is measured using a power meter (Rohde & Schwarz NRP2) connected through a GSG 110-GHz probe. Cable, bias-tee and probe losses are calibrated with a vector network analyser and appropriate calibration kits (85059B and CS-5). For frequencies above 110 GHz, a THz power meter (VDI PM5B) and corresponding waveguide probes (110–325 GHz) are used. Losses from waveguide tapers and probes are de-embedded using data provided by the manufacturer.

$$P_{\text{ideal}} = \frac{1}{2} R_L I_{\text{ph}}^2 \tag{6}$$

**Details of data transmission experiments**
For short-reach IMDD experiments, periodic pseudorandom bit sequence signals are generated by an AWG (Keysight M8199B, approximately 75-GHz analogue bandwidth) and shaped with a raised cosine filter. An electrical power amplifier (SHF T850 C, 100-GHz analogue bandwidth) is needed to compensate for the degradation brought by the cables and probes. After amplification, the TFLN modulator encodes the 1,550-nm optical carrier into NRZ and PAM-4 formats at symbol rates from 112 to 256 Gbaud. An EDFA (Amonics AEDFA-C-DWDM) is used to boost the modulated optical signal. At the Rx side, the modulated signals are first converted and recorded by a DSO (Keysight N1000A) with a 120-GHz optical sampling module (Keysight N1032A). We apply no bandwidth compensation DSP and take screenshots of the eye diagrams to show the intrinsic UWB features of our TFLN modulator. The transmission capacity can be further improved using feed-forward equalizer and other equalization algorithms. We also use a 70-GHz-bandwidth RTO (Keysight UXR0702AP) with a 110-GHz PD (Coherent XPDV4121R) to obtain the real transmission data. After resampling and synchronization, the collected data are fed into the network for signal recovery. Finally, the output signal from the complex-biGRU algorithm undergoes signal decision-making and BERs are calculated to assess the system performance.

For high-speed wireless coherent optical transparent relaying demonstration, the optical baseband signal is generated by a commercial silicon coherent transmitter with a bandwidth of around 35 GHz and a laser source operating at the fixed wavelength of 1,550 nm. IQ signals of different symbol rates are generated by an AWG and mapped to QPSK, 16-QAM and 32-QAM symbols. A tunable ECL (TOPTICA CTL 1550) with 180-GHz frequency offset from 1,550 nm serves as the Tx LO, which is attenuated to the same power as the signal light and combined with the signal light by means of a 50:50 coupler. The mixed signals are amplified by an EDFA and coupled into the modified UTC-PD, in which the baseband signal is shifted to the 180-GHz-centred THz signal by heterodyne beating. Two 140–220-GHz horn antennas (Anteral SGH-26-WR05) with 26-dBi gain are placed at a distance of 20 cm for THz emitting and receiving. A 145–220-GHz LNA (Fintest FTHZLNA-05FB) with 24-dB gain is used at the output of the Rx antenna to amplify the attenuated THz signal, which subsequently drives the TFLN modulator and converts onto an optical carrier with the wavelength of 1,550 nm. Before EDFA amplification, the optical carrier is filtered out by a tunable FBG filter (AOS Tunable FBG) to take full advantage of the available gain of the EDFA. The output from the EDFA is sent into the optical modulation analyser (Keysight N4391B) as the signal light. Another tunable ECL acting as the Rx LO is set to be the same wavelength as the Tx LO. The data are then collected and analysed offline using different DSP techniques. For 4-m wireless transmission, we replace the original horn antennas with 40-dBi-gain lens antennas while keeping the rest of the link unchanged.

In the baseline DSP chain at the receiver, Gram–Schmidt orthogonal normalization is first performed, followed by matched filtering and downsampling. Adaptive equalization and carrier recovery are then implemented. Subsequently, frequency offset is estimated, in which the tap coefficients of the equalizer are pre-converged using the constant modulus algorithm. Afterwards, carrier phase estimation is performed within the decision-directed least mean square update loop using the blind phase search algorithm. Finally, another orthogonalization is applied. For the complex-biGRU processing chain, the equalized signal from baseline DSP is used as the input signal and fed into the GRU network for further equalization. Symbol decisions are determined and BERs are calculated after each DSP chain.

For multichannel real-time videos transmission, two Ethernet-optical switches (FS S3950-4T12S-R) are used for signal routing and distribution. The computers connect to the Tx/Rx switch through Ethernet ports under the IEEE 802.3ab protocol. Owing to the fixed 10-GHz bandwidth limitation of the Rx FBG filter, the carrier spacing between two simultaneously transmitted video signals must exceed 10 GHz to avoid signal aliasing. Therefore, we select two SFP+ modules (FC DPP1-5592-81Y1) with 17-GHz wavelength spacing as the transmitters.

**Details of the complex-biGRU algorithm**
Wireless communication, especially higher-order wireless coherent communication, sustains more severe nonlinear effects caused by reflection, diffraction and scattering in the free space. Various nonlinear equalizers have been proposed to mitigate these repercussions[32,37], among which NN-based methods outperform others on dealing with complicated disturbance. However, although NN-based equalizers have demonstrated effectiveness in reducing the BER, the 'jail window' pattern often arises simultaneously (Supplementary Note 5). Such phenomenon will change the signal distribution away from a Gaussian distribution, leading to serious deterioration of FEC coding[51].

The proposed complex-biGRU network implements a five-layer structure. The first layer is the input layer. For coherent optical communication, each symbol $r_n$ can be separated into $I_n$ and $Q_n$. The second layer is the complex-biGRU layer. Given that the M-QAM modulation format encompasses both in-phase (I) data and quadrature (Q) data, the complex-biGRU layer is designed to process the I and Q data in parallel. In each biGRU network, the model with the bidirectional structure is determined by the combined states of two GRUs, which are unidirectional in opposite directions. One GRU processes the data from the start of the sequence and the other GRU processes the data from the end of the sequence. The outputs of the complex-biGRU layer are fully connected to a linear layer, followed by a multilevel layer serving as the nonlinear activation function. Finally, the predicted output can be expressed as the summation of the two-path outcomes. For IMDD signals, only one path of the network is activated to reduce computational expenditure of the system, whereas the remaining processes are the same as those for coherent signals.

Traditional activation functions typically exhibit two saturation regions. Nevertheless, for higher-order modulation formats, the equalizer equipped with an activation function featuring two saturation regions demonstrates insufficient effectiveness. The signal equalization problem can be considered as a classification problem, in which the output of a multilevel signal needs to be categorized into several classes. Therefore, an activation function with several saturation regions will greatly boost the performance of the system. The activation function applied in our algorithm is defined as[52]:

$$f(x) = \frac{2\alpha_2}{1 + e^{-\alpha_1(x - 2\mu)}} - \alpha_2 + 2\mu\,\alpha_2 = \frac{1 + e^{-\alpha_1}}{1 - e^{-\alpha_1}} \tag{7}$$

in which $\alpha_1$ represents the gradient factor and $\alpha_2$ is devised to ensure the continuity of the function. For four-level signal equalization scenarios such as PAM-4 and 16-QAM, $\mu$ assumes values of −1, 0 and 1 when $x \le -1$, $-1 < x \le 1$ and $x > 1$, respectively. Similarly, to accommodate PAM-6 and 32-QAM signals, $\mu$ can be adjusted to −2, −1, 0, 1 and 2 when $x \le -3$, $-3 < x \le -1$, $-1 < x \le 1$, $1 < x \le 3$ and $x > 3$, respectively.

## Comparison of the UWB integrated photonics system

The proposed UWB integrated photonics system achieves exceptional performance for both the device and system levels. To be more specific, we summarize the representative results in recent years and conduct detailed comparison analyses with them. Extended Data Table 1 shows the comparison of integrated EO Mach–Zehnder modulators (MZMs) operating in the C band on different material platforms with different structures. Our modulator achieves double the bandwidth compared with most other devices while maintaining high modulation efficiency and ultralow insertion loss. We also calculate the slope efficiency of the modulator using following equation:

$$\text{Slope efficiency} = 20 \times \log\left(\frac{T}{V_{\pi,\text{RF}}}\right) \tag{8}$$

in which $T$ is the optical transmittance before and after the modulator in linear units. Our modulator realizes comparable slope efficiency while operating at twice the bandwidth. Although plasmonic modulators have demonstrated nearly 1-THz EO bandwidth, they simultaneously suffer from high optical insertion loss and prohibitively large half-wave voltage, as well as fluctuating EO response, limiting their performance in practical applications.

Extended Data Table 2 presents a comparison of integrated high-speed OE photodetectors across different material platforms and epitaxial structures. At the material level, although Ge-based PDs have achieved impressive bandwidths (for example, 265 GHz), they typically operate under low optical power owing to the limited electron saturation velocity in Ge (about $5 \times 10^6$ cm s$^{-1}$), which restricts current handling at high frequencies and application for THz generation. By contrast, InP offers a higher electron saturation velocity (about $2 \times 10^7$ cm s$^{-1}$), enabling better current handling capability, making it the preferred material for high-speed, high-power photodetectors beyond 100 GHz. From a structural perspective, UTC-PDs exhibit superior saturation characteristics over conventional PIN PDs by rapidly collecting slow-mobility holes and enabling electron-only drift. This design mitigates space-charge effects under high optical injection, thus supporting higher photocurrents and THz output power. Among the InP-based waveguide-coupled PDs, our modified UTC-PD exhibits a higher OE bandwidth than previous works while maintaining high THz output power, along with a decent responsivity of 0.24 A W$^{-1}$. These well-balanced metrics position it as a compelling solution for next-generation ultra-broadband telecommunication systems.

Extended Data Table 3 shows the comparison of the integrated IMDD communications. Our approach demonstrates the highest transmission rate without any bandwidth compensation DSP. We also achieve the highest symbol rate of 256 Gbaud and the highest PAM-4 data rate of 512 Gbps with the complex-biGRU algorithm, reaching the limit of our test system. On the basis of these results, 479 Gbps (=512/(1 + 0.07)) net bit rate using conventional PAM-4 modulation can be calculated, achieving performance comparable with advanced PS-PAM-16 formats.

Extended Data Table 4 shows the comparison of the high-speed THz wireless communications based on different approaches. Our system is prominent in both single-channel symbol rate and single-channel data rate using the complex-biGRU algorithm. Even with the baseline DSP method, our system still achieves the highest single-channel symbol rate and data rate. We also calculate the carrier utilization efficiency, defined as the single-channel data rate divided by the carrier frequency, to evaluate the information-carrying ability of the carrier without any multiplexing techniques. Theoretically, a higher carrier frequency enables larger transmission capacity. Our system achieves the highest carrier utilization efficiency of 2.22, tripling the performance of previous systems. Besides single-channel performance, we also investigate multichannel capabilities across different THz wireless communications methods. As shown in the last column, the number of channels is used to characterize the multichannel performance of the system. Our solution achieves the maximum number of transmission channels of 86 channels with 1-GHz channel bandwidth, demonstrating exceptional large-density-access performance.

## Data availability

The data used to produce the plots in this paper are available from Zenodo at https://doi.org/10.5281/zenodo.18168790 (ref. 53).

## Code availability

The codes that support the findings of this study are available from the corresponding authors.

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

**Acknowledgements** This work is supported by the National Key Research and Development Program of China (grant no. 2022YFB2803700) and the National Natural Science Foundation of China (grant nos. 62322501, 62235002 and 623278110). We gratefully acknowledge the device fabrication support from the ShanghaiTech University Materials Device Lab (SMDL) and also acknowledge that part of the experimental work is supported by the Core Facility Platform of Electronics, School of Information Science and Technology (SIST), ShanghaiTech University. We thank Q. Wu and W. Feng for fruitful discussions and Q. Xie, C. Li and P. Yan for the guidance on the use of the instruments.

**Author contributions** The experiments were conceived by Y.Z., Y.G. and H.S. The TFLN modulator was designed and characterized by Y.Z., P.Z., J.C., L. Yao and X.X. The InP-based UTC-PD was designed and characterized by Luyu Wang, L.L. and Z.Z. and packaged by T.L. The AI algorithm was conceived by Y.Z., Y.S. and J.Q. Short-distance transmission experiments were conducted by Y.Z., Y.G., L. Yang and K.L., with the assistance of Y.T., Z.X., Lei Wang and C.H. Long-distance transmission experiments were conducted by Y.Z., Y.G., L. Yang, with the assistance of Z.H. The results were analysed by Y.Z. and Y.G. All authors participated in writing the manuscript. The project was under the supervision of H.S., B.C., S.Y. and X.W.

**Competing interests** Luyu Wang, L.L., T.L. and B.C. are involved in developing uni-travelling carrier photodiodes technologies at FastPhide Photonics Ltd. The other authors declare no competing interests.

**Additional information**
**Correspondence and requests for materials** should be addressed to Haowen Shu, Baile Chen, Shaohua Yu or Xingjun Wang.

**Extended Data Table 1 | Comparison of the integrated modulators[54]**

| Modulator structure | EO Bandwidth (GHz) | Length (mm) | Insertion loss (dB) | $V_{\pi,LF}$ (V) | Slope efficiency (dB) | Symbol rate (Gbaud) |
|---|---|---|---|---|---|---|
| **TFLN + Slow-wave** | **220** | **5** | **0.6** | **5.1** | **-17.85 @ 220 GHz** | **256** |
| TFLN + Slow-wave [54] | 110 | 23.5 | N/A | 1 | N/A | 130 |
| TFLN + Equalization [29] | 110 | 45 | 3.2 | 2.4 | -14.34 @ 110 GHz | N/A |
| TFLN [50] | 170[a] | 5.8 | N/A | 4.6 | N/A | N/A |
| Si + Equalization [30] | 110 | 0.9 | 4.3 | 54 | -44.25 @ 110 GHz | 140 |
| Si + Slow-light [10] | 110 | 0.124 | 6.8 | 78 | -54.44 @ 110 GHz | 112 |
| Plasmonic [15] | 997 | 0.01 | 5.6 | 23.4 | -41.58 @ 997 GHz | N/A |
| InP [11] | 100 | 3.6 | 3.5 | 1.5 | -13.52 @ 100 GHz | 200 |
| TFLT [13] | 110 | 6 | 0.23[b] | 4.8 | -17.09 @ 110 GHz[b] | 208 |

[a]Estimated from the measured electrical parameters. [b]Estimated from the simulated results.

Extended Data Table 2 | Comparison of the integrated PDs[55–59]

| Photodiode structure | OE Bandwidth (GHz) | Saturation power (dBm) | External responsivity (A/W) | Dark current (nA) |
|---|---|---|---|---|
| **InP-MUTC** | **250** | **1.26 @ 140 GHz** | **0.24** | **0.1 (-5 V)** |
| InP-MUTC [16] | 230 | -4.94 @ 220 GHz | 0.07 | 11.5 (-2 V) |
| InP-MUTC [55] | 156 | 0.69 @ 150 GHz | 0.165 | 12.6 (-3 V) |
| InP-MUTC [56] | 153 | -5.6 @ 130 GHz | 0.38 | 3 (-2 V) |
| InP-NB-UTC [18] | 220 | 0.33 @ 165 GHz | 0.11 | N/A |
| InP-UTC [57] | 170 | -2 @ 150 GHz | 0.27 | N/A |
| InP-PIN [58] | 130 | -8.3 @ 130 GHz | 0.5 | 2 (-5 V) |
| Ge-PIN [59] | 265 | N/A | 0.12 | 200 (-2 V) |

**Extended Data Table 3 | Comparison of the integrated IMDD fibre communication[60-63]**

| Modulator structure | Single lane data rate | | | Single lane symbol rate | | Algorithm |
|---|---|---|---|---|---|---|
| | Format | Net rate (Gbps) | Rate (Gbps) | Format | Rate (Gbaud) | |
| **TFLN + Slow-wave** | **PAM-4** | **479** | **512** | **NRZ** | **256** | **Complex-biGRU** |
| | **PAM-4** | **N/A** | **392** | **NRZ** | **210** | **DSP-free** |
| TFLN + Equalization [12] | PAM-8 | 440 | 528 | NRZ | 255 | 51-tap DFE |
| TFLN + Slow-wave [60] | PAM-4 | 303 | 364 | NRZ | 196 | DSP-free |
| TFLN [61] | PS-PAM-16 | 494 | 660 | PS-PAM-16 | 200 | 2201-tap LMS |
| Si + Equalization [30] | NRZ | N/A | 140 | NRZ | 140 | DSP-free |
| Si + Slow-light [62] | PAM-4 | 374 | 400 | PAM-4 | 200 | biGRU |
| Si [63] | PAM-4 | 287 | 308 | NRZ | 182 | 32-tap FFE |
| TFLT [13] | PAM-8 | 405 | 528 | NRZ | 208 | DD-LMS |
| Plasmonic [14] | PAM-8 | 363 | 432 | NRZ | 205 | 301-tap FFE + 7-tap DFE |

**Extended Data Table 4 | Comparison of high-speed THz wireless communication**

| Method | Carrier frequency (GHz) | Carrier utilization efficiency (bit/s/Hz) | Distance (m) | Single channel data rate | | Single channel symbol rate | | Multi-channel capability |
|---|---|---|---|---|---|---|---|---|
| | | | | Format | Rate (Gbps) | Format | Rate (Gbaud) | |
| All-optical | 180 | 2.222 | 0.2[a] | 32-QAM | 400 | 16-QAM | 100 | 86 |
| | | 1.333 | 0.2[a] | 16-QAM | 240 | QPSK | 90 | |
| | | 2.222 | 4[b] | 16-QAM | 400 | QPSK | 100 | |
| | | 1.689 | 4[b] | 16-QAM | 304 | QPSK | 96 | |
| | 101 [25] | 0.707 | 20 | 64-QAM-OFDM | 71.4 | 32-QAM-OFDM | 11.9 | 1 |
| | 231 [26] | 0.103 | 5 | 10-Nyquist-FDM | 24 | 10-Nyquist-FDM | 7.5 | 10 |
| | 288.5 [2] | 0.173 | 16 | QPSK | 50 | QPSK | 25 | 1 |
| All-electric | 220 [19] | 0.095 | 1260 | 16-QAM-OFDM | 21 | 16-QAM-OFDM | 5.25 | 4 |
| | 237.5 [20] | 0.421 | 20 | 16-QAM | 100 | 16-QAM | 25 | 3 |
| | 465 [21] | 0.563 | 10 | PS-64-QAM | 262 | PS-64-QAM | 46 | 1 |
| Hybrid optoelectronic | 100 [22] | 0.640 | 1 | 16-QAM | 64 | 16-QAM | 16 | 1 |
| | 250 [23] | 0.200 | 0.1 | 16-QAM | 50 | 16-QAM | 12.5 | 1 |
| | 340 [24] | 0.353 | 3 | 8-QAM | 120 | 8-QAM | 40 | 2 |

[a]Conducted with horn antennas. [b]Conducted with lens antennas.