## [Peer Review file · Nature]

Integrated photonics enabling ultra-wideband fiber-wireless communication

Corresponding Author: Professor Xingjun Wang

Version 1:

Reviewer comments:

Referee #1

(Remarks to the Author)

The authors presented a very impressive work to propose and demonstrate PIC-based, ultra-broadband radio–optical systems to span microwave to mmWave/THz for next-gen networks. They pushed absolute bandwidth & rate with a >220–250 GHz EO/OE chain (TFLN modulator + modified UTC-PD) and achieved 512 Gb/s fiber and 400 Gb/s THz (\approx 180 GHz) wireless, plus 86-channel 8K video from 138–223 GHz using a complex-biGRU equalizer. Multiple world records have been achieved in this work. The manuscript can be considered for acceptance after addressing comments/questions below:

1. For TFLN modulator bandwidth beyond 110 GHz in in Fig. 2(c), how exactly were sideband amplitudes converted to $|S_{21}|$? What calibration/de-embedding was applied? It would better to include uncertainty bars vs. frequency.
2. Please include the S_{11} data in Fig. 2(c).
3. For the >200 GHz bandwidth claim, did authors define 3 dB bandwidth relative to low-frequency gain or to the best in-band level? Please clarify drive voltage, $V_{\pi-L}$, and electrical launch conditions at a few anchor frequencies, e.g., 40/110/180/220 GHz.
4. Why do S_{21} data points around 100GHz scatter a lot in Fig. 2(f)? Similarly, it would better to discuss the device performance consistency/uncertainty a bit more.
5. What's the UTC PD responsivity? What are the bandwidth, saturation power, dark current and linear output IP3 of UTC-PD at high-temperature, say 100 C?
6. Were any averaging and filtering used in eye diagrams of Fig. 2(h)?
7. In Fig. 2(i), BER is shown with complex-biGRU post-processing. Any EDFA noise loading? Please discuss equalizer tap counts/complexity used to reach the HD-FEC threshold.
8. What would be the raw BER without the complex-biGRU? How much latency and power it would incur to run the complex-biGRU?
9. I wouldn't say 6.2 V of V_{pi-RF} low.
10. For the complex-biGRU, please specify model size, training data volume, compute time, and inference latency; compare against practical real-time budgets.
11. In the 180 GHz wireless experiments, what RF power reached the transmit horn, and what were the antenna gains, cable losses? Provide a complete equivalent isotropically radiated power and receiver sensitivity budget.

12. How are the 86 channels generated and multiplexed (optically/electrically)? What guard bands, adjacent-channel isolation, and aggregate EVM/BER did you achieve across 138–223 GHz?
13. Was the video demo truly simultaneous across 86 channels, or time-multiplexed? Please provide aggregate throughput and power.
14. Since many optical and RF amplifiers were used in this demo, please elaborate the system energy efficiency. Please provide a full power/energy budget (lasers, modulators/drive, UTC-PD bias, LNAs, DSP) and throughput/W figures for both fiber and wireless modes.
15. What is the packaging scheme to integrate TFLN EO modulator and InP UTC-PD devices with fiber arrays and THz antennas while keeping the >200 GHz performance? In particular, please outline thermal management and connector constraints for ultra-high speed operation.
16. The architecture uses all-photonic up/down-conversion with mirrored Tx/Rx OEO-derived LOs. Please quantify LO frequency-set accuracy, zero-IF alignment error, and capture range under temperature drift.
17. "We also anticipate more ...in monolithically fully integrated optoelectronic systems". Please elaborate how to monolithically integrate lasers and other components on the TFLN platform?

Referee #2

(Remarks to the Author)

The manuscript titled "Integrated photonics enabling ultra-wideband fiber–wireless communication" reports a seamless data path between optical fiber and wireless channels. The key components are thin-film lithium niobate modulators and uni-traveling-carrier photodetectors. The lithium niobate modulator achieves a record-high 3-dB bandwidth of 260 GHz, enabling a record-high single-channel data rate. When combined with a custom-designed unitraveling-carrier photodiode, the authors demonstrate record-fast fiber–wireless–fiber data communication.

While the results advance the state of the art in terms of performance metrics, the contribution appears to be primarily an engineering refinement rather than a conceptual or scientific breakthrough. The modulator and photodiode designs largely follow established approaches [Ref. 27], and the wireless link demonstrated is limited to a 20-cm distance, which raises questions about its relevance to the envisioned applications. Moreover, it is unclear how the reported 8 mA photocurrent corresponds to 0 dBm of RF output power—clarification is needed on whether this represents the actual radiated power from the antenna.

Although the authors measured an impressive 3-dB bandwidth of 260 GHz, the optical–wireless link ultimately operates at ~100 Gbaud. This regime has been extensively explored in prior works across multiple platforms (e.g., Schuh et al., OFC Th5B.5; Lin et al., Opt. Express 27, 5610 (2019); Xu et al., Nat. Commun. 11, 3911 (2020)). While those studies primarily focused on fiber communication, the present manuscript does not convincingly demonstrate that the wireless integration offers a significant practical advantage. The applicability of the wireless demonstration to real-world scenarios remains uncertain.

It is no doubt that the authors have a good device and got some really nice data. However, I do not think the level of innovation or the performance of the device warrants the paper's publication in a high-impact journal like Nature.

Referee #3

(Remarks to the Author)

The authors present key integrated photonic components — specifically, an electro-optic (E/O) modulator and a photodetector — incorporated into a complete system that supports an impressive analog bandwidth exceeding 250 GHz. While previous works have demonstrated either plasmonic modulators or Ge-PIN photodetectors with even higher total electrical bandwidths (notably in Refs. [13] and [60]), those results were achieved using different technologies with inherent limitations. In contrast, the authors clearly outperform other related works in terms of both bandwidth and single-lane (channel) data rate for IMDD DSP-free, and all-THz communications.

The practical applicability of the developed system is convincingly demonstrated by the successful transmission of 86 channels carrying 8K video streams across a spectral range from 138 GHz to 223 GHz.

I have one question regarding the frequency stability of the resulting signal. Since a heterodyne up-conversion method was used both for the E/O frequency response measurement and for the up-conversion in the system demonstration, could the authors comment on the stability of the beat signal, considering that two independent lasers were used?

Overall, the manuscript is well organized and meets all formal requirements. In my opinion, it makes a significant contribution to the advancement of converged optical/THz communication systems by demonstrating high-performance integrated components that hold promise for next-generation mobile networks, data centers, and beyond. Furthermore, the level of novelty is sufficient within the context of ultra-wideband microwave photonics systems. Therefore, I recommend the manuscript for publication.

Version 2:

Reviewer comments:

Referee #2

(Remarks to the Author)

While I still think that the results shown in this work are mainly engineering refinement, I agree with the authors that the full-system demonstration is heroic and record-setting. In this sense, I agree that this work can be published in Nature.

I just have one more minor question about the data. The EO and OE responses shown in Fig. 2 (c) and (f) are somewhat wiggly. I believe these are real features of the system rather than measurement artifacts. Are these features contributing to the raw error rate currently measured? If so, is the complex-biGRU algorithm effective at correcting for this error?

Referee #3

(Remarks to the Author)

In my opinion, the authors have adequately responded to the reviewers' comments, and I recommend the manuscript for publication.

We appreciate the careful review by the reviewers and have modified the manuscript in accordance with their suggestions. Here, we present a point-by-point reply (in blue) to the reviewers' comments (in black), as well as the action taken (in red).

Response to the report from the Referee #1

General comments: “The authors presented a very impressive work to propose and demonstrate PIC-based, ultra-broadband radio–optical systems to span microwave to mmWave/THz for next-gen networks. They pushed absolute bandwidth & rate with a >220–250 GHz EO/OE chain (TFLN modulator + modified UTC-PD) and achieved 512 Gb/s fiber and 400 Gb/s THz (≈ 180 GHz) wireless, plus 86-channel 8K video from 138-223 GHz using a complex-biGRU equalizer. Multiple world records have been achieved in this work. The manuscript can be considered for acceptance after addressing comments/questions below:”

Our reply: We appreciate the reviewer’s recognition of our work and sincerely thank the reviewer for the thorough evaluation of our manuscript and constructive feedback. Additional experiments and analysis have been conducted to elaborate the motivation and innovation of our work. The comments and our point-by-point responses are provided below.

Comment 1: “For TFLN modulator bandwidth beyond 110 GHz in Fig. 2(c), how exactly were sideband amplitudes converted to $|S_{21}|$? What calibration/de-embedding was applied? It would better to include uncertainty bars vs. frequency.”

Our reply: We appreciate the reviewer’s comment, which allows us to perform a detailed explanation of the modulator bandwidth measurements beyond 110 GHz. We use optical spectrum analyzer (OSA) to measure and calculate the EO response beyond 110 GHz, which is widely adopted as the standard method for measuring modulator bandwidth in terahertz frequency band [1-3]. As we described in the Methods of the manuscript, the electro-optic response at each frequency can be derived as:

$$P_n = \frac{P_{\text{carrier}}}{P_{\text{sideband}}} \quad (\text{R1})$$

$$V_{\pi,\text{RF}} = V_{\pi,\text{LF}} \times 10^{\frac{EO S_{21}}{20}} \quad (\text{R2})$$

$$V_{\pi,\text{RF}} = \frac{1}{4} \pi V_p \sqrt{P_n} \quad (\text{R3})$$

$$EO S_{21} = -20 \times \lg \left(\frac{\pi V_p}{4V_{\pi,\text{LF}}} \right) - 10 \times \lg(P_n) \quad (\text{R4})$$

This equation can be further transformed and decomposed as:

$$EO S_{21} \text{ (dB)} = C - P_m - P_c + P_s \quad (\text{R5})$$

where $C = -20 \times \lg \left(\frac{\pi}{4V_{\pi,\text{LF}}} \right)$ represents the constant term related to low radio frequency half-wave voltage ($V_{\pi,\text{LF}}$). $P_m = 20 \times \lg(V_p)$ is the drive power delivered to the modulator at each test frequency. P_c and P_s are the carrier and sideband power measured from the OSA corresponding to each test point. Since the probe is directly connected to the frequency multiplier without any waveguides or cables, the driving power can be calculated by subtracting the probe loss from the output power of the frequency multipliers measured directly by a RF power meter. Figure.R1 shows the calibrated driving power from 110 GHz to 170 GHz and from 140 GHz to 220 GHz. **By de-embedding the difference of the driving**

power, from the Eq.R5 one can tell that the EO response of the modulator over 110 GHz varies only with the carrier and sideband amplitudes.

Figure.R1 Driving power for (a) 110 GHz–170 GHz and (b) 140 GHz–220GHz.

During the measurement process, we use MATLAB to control the microwave source for generating different frequencies (110-170 GHz & 140-220 GHz) and collect the corresponding modulator output optical spectra. For data analyzing, we extract the amplitudes of the carrier and sidebands and calculate the EO response within each test band according to Eq.R5. P_n is determined by averaging the upper and lower sidebands to mitigate measurement errors. The calculated results for 110-170 GHz and 140-220 GHz bands are first normalized to the lowest by their overlapping-region average and finally referenced to the 110 GHz data from LCA. **Since the probes and frequency multipliers have been precisely measured and calibrated, the primary source of measurement uncertainty lies in the power uncertainty of the OSA. According to the operation manual, the power measurement uncertainty of the OSA is ± 0.4 dBm, resulting in a bandwidth measurement uncertainty of ± 0.4 dB.**

To more clearly demonstrate the conversion process from spectral sidebands to bandwidth and the uncertainty of our measurements, we have revised the manuscript as follows:

(Main part, Page 4 Figure 2)

Figure.2 (b) Optical spectrum of the modulated sidebands (carrier omitted). **The bandwidth measurement uncertainty from 110 GHz to 220 GHz is ± 0.4 dB due to the power uncertainty of OSA.**

(Methods, Characterization of the EO/OE response of TFLN modulator and modified UTC-PD) The electro-optic response can be derived from the Eq.1-3 as:

$$EO S_{21} = -20 \times \lg \left(\frac{\pi}{4V_{\pi,LF}} \right) - 20 \times \lg(V_p) - P_c + P_s \quad (4)$$

The first term of the Eq.4 represents the constant term related to $V_{\pi,LF}$. The second term is the drive power delivered to the modulator and can be calculated by subtracting the probe loss from the output power of the frequency multipliers. P_c and P_s are the carrier and sideband power measured from the OSA corresponding to each test point. After de-embedding the difference of the driving power at each test frequency, the EO response is proportional only to the carrier and sideband amplitudes. Since the probes and frequency multipliers have been precisely measured and calibrated, the primary source of measurement uncertainty lies in the power uncertainty of the OSA. According to the operation manual, the power measurement uncertainty of the OSA is ± 0.4 dBm, resulting in a bandwidth measurement uncertainty of ± 0.4 dB from 110 GHz to 220 GHz.

Reference:

- [1] Zhang, Y. et al. Systematic investigation of millimeter-wave optical modulation performance in thin-film lithium niobate. *Photonics Research* 10, 2380–2387 (2022).
- [2] Mercante, A. J. et al. Thin film lithium niobate electro-optic modulator with terahertz operating bandwidth. *Optics express* 26(11), 14810-14816 (2018).
- [3] Horst, Y. et al. Ultra-wideband MHz to THz plasmonic EO modulator. *Optica* 12, 325–328 (2025).

Comment 2: “Please include the S11 data in Fig. 2(c).”

Our reply: We agree with the reviewer that the S11 parameter serves as a critical metric for characterizing microwave reflection of the modulator electrodes. As illustrated in Figure.R2, the electrical reflection S11 remains mostly below –10 dB within 0-110 GHz, which indicates its impedance matching performance. For frequencies beyond 110 GHz, the S11 parameter cannot be directly measured due to the limitations of the test equipment. However, the EO S21 curve reveals a gradual decline without significant periodic resonances, indicating that the modulator’s electrodes also maintain excellent impedance matching across 110-220 GHz.

Figure.R2 Measured EO response and electrical reflection from 1 GHz to 110 GHz.

Additional descriptions about the modulator electrodes’ electrical reflection are therefore added as follows:

(Supplementary Note I) The measured EO response from 0-110 GHz is depicted in Figure.S3. The EO S21 exhibits a flat response up to 110 GHz, while the electrical reflection S11 remains mostly below –10 dB, which indicates its impedance matching performance. For frequencies over 110 GHz, we can observe a gradual decline of the EO S21 without significant periodic resonances, indicating that the modulator’s electrodes also maintain excellent impedance matching across 110-220 GHz.

Figure.S3 Measured EO response and electrical reflection from 1 GHz to 110 GHz.

Comment 3: “For the >200 GHz bandwidth claim, did authors define 3 dB bandwidth relative to low-frequency gain or to the best in-band level? Please clarify drive voltage, $V_{\pi-L}$, and electrical launch conditions at a few anchor frequencies, e.g., 40/110/180/220 GHz.”

Our reply: We define 3 dB bandwidth relative to low-frequency gain (i.e., 1 GHz point), which has been commonly adopted for the characterization of EO modulator with ultra-wideband bandwidth over 110 GHz [1,2]. Thanks to the excellent impedance matching design, our modulator also obtains a flat EO response with the maximum deviation of 0.5 dB above 0 dB level. Even referring to the best in-band level, our modulator still demonstrates an EO bandwidth exceeding 200 GHz.

An overview of the experimental setup across the whole frequency bands is also provided. For 0-110 GHz, a coaxial cable with 1.0 mm connector is employed to load the signal. The drive power for S21 characterization can be calculated by subtracting the probe and cable losses from the output power of the LCA, while the drive voltage is derived from the drive power with 50-Ω impedance (Figure.R3a). For 110-170 GHz and 140-220 GHz, WR-6 and WR-5 rectangular waveguide are employed to deliver the corresponding frequency signal to the modulator. The drive voltage can also be derived from the drive power (Figure.R1) with 50-Ω impedance, as shown in Figure.R3b and Figure.R3c. Based on the measured EO response and $V_{\pi,LF}$, $V_{\pi,RF}$ from 1-220 GHz can be calculated according to Eq.2 and $V_{\pi}\cdot L$ at each frequency can be obtained by simply multiplying the modulator length of 5 mm (Figure.R4).

Figure.R3 Drive voltage for S21 characterization at the bands of (a) 0-110 GHz, (b) 110-170 GHz, and (c) 140-220 GHz.

Figure.R4 $V_{\pi}\cdot L$ of the ultra-wideband TFLN modulator.

To clarify the definition of the measured 3 dB bandwidth, we have revised the relevant description in the manuscript. We have also added contents about bandwidth measurements conditions and $V_{\pi}\cdot L$ in the supplementary information:

(Supplementary Note I) After device fabrication, we analyze the EO response and impedance matching performance of the modulators. An overview of the experimental setup across the whole frequency

bands is also provided. For 0-110 GHz, a coaxial cable with 1.0 mm connector is employed to load the signal. The drive power for S21 characterization can be calculated by subtracting the probe and cable losses from the output power of the LCA, while the drive voltage is derived from the drive power with 50 Ω impedance (Figure.S2a). For 110-170 GHz and 140-220 GHz, WR-6 and WR-5 rectangular waveguide are employed to deliver the corresponding frequency signal to the modulator. The drive voltage can also be derived from the drive power with 50 Ω impedance, as shown in Figure.S2b and Figure.S2c.

Figure.S2 Drive voltage for S21 characterization at the bands of (a) 0-110 GHz, (b) 110-170 GHz, and (c) 140-220 GHz.

We also measure the half-wave voltage of the modulator using 100 kHz triangular sweep as shown in Figure.S4. We obtain a $V_{\pi,LF}$ of 5.1V. Based on the measured EO response and $V_{\pi,LF}$, $V_{\pi,RF}$ from 1-220 GHz can be calculated according to Eq.2 and $V_{\pi} \cdot L$ at each frequency can be obtained by simply multiplying the modulator length of 5 mm (Figure.S5).

Figure.S5 $V_{\pi} \cdot L$ of the ultra-wideband TFLN modulator.

(Main part, Page 3 line 53) The whole EO response shows an ultrahigh experimental bandwidth of >220 GHz (Fig.2c) and an extrapolated result of about 260 GHz (with respect to 1 GHz).

Reference:

- [1] Kharel, P. et al. Breaking voltage-bandwidth limits in integrated lithium niobate modulators using micro-structured electrodes. *Optica* 8, 357–363 (2021).
- [2] Zhang, Y. et al. Systematic investigation of millimeter-wave optical modulation performance in thin-film lithium niobate. *Photonics Research* 10, 2380–2387 (2022).

Comment 4: “Why do S21 data points around 100GHz scatter a lot in Fig. 2(f)? Similarly, it would better to discuss the device performance consistency/uncertainty a bit more.”

Our reply: We thank the reviewer for pointing out this concern. **The fluctuations observed in the S21 curve, particularly around 100 GHz, arise from RF return loss at the GSG probe-tip plane due to the non-ideal 50-Ω impedance at the probe port.** Our previous calibration followed the conventional measurement approach used for high-speed photodiodes [1,2], which treated the probe port as an ideal 50-Ω termination and only accounted for the insertion loss of the probe, i.e., using the formula shown below:

$$P'_L = P_L \frac{1}{|S_{\text{probe},21}|^2} \quad (\text{R6})$$

P'_L denotes the ideal output power generated by the PD chip, whereas P_L refers to the power actually measured by the collection system. In practice, the probe exhibits a complex impedance that introduces both transmission loss and partial signal reflection along the measurement path, resulting in fluctuation in the measured response. Therefore, to accurately determine the PD’s intrinsic output power, it is essential to precisely characterize the probe scattering parameters ($S_{\text{probe},11}$ and $S_{\text{probe},21}$), as well as the reflection coefficient of the PD (Γ_{PD}). Accordingly, the calibration formula is refined as [3]:

$$P'_L = P_L \frac{|1 - \Gamma_{\text{PD}} S_{\text{probe},11}|^2}{|S_{\text{probe},21}|^2} \quad (\text{R7})$$

In our improved calibration procedure, the probe’s scattering matrix S_{probe} is first measured and de-embedded using a 110 GHz vector network analyzer (VNA). Then, the intrinsic S11 of the UTC-PD is measured with the calibrated probes and Γ_{PD} can be calculated. By applying the updated Eq.R7, the PD’s intrinsic output power and frequency response are accurately determined, independent of the external measurement system. As shown in Figure.R5, compared to the conventional method (Figure.R5a), the improved calibration procedure (Figure.R5b) yields a substantially smoother frequency response curve with the scattering around 100 GHz effectively eliminated.

Figure.R5 (a) Measured optic-electro S21 curve of the UTC-PD by conventional method. (b) Re-calibrated S21 curve from (a) by considering impedance compensation. (c) RSD of the ten repeated measurement results across DC–110 GHz.

We further evaluated measurement consistency by performing ten repeated heterodyne sweeps under identical optical power and bias conditions. The resulting relative standard deviation (RSD) of the

improved S21 amplitude remains below 2 % across most of the frequency points in DC–110 GHz band, as shown in Figure.R5c. This indicates that the overall system uncertainty is low and the measured response is highly repeatable, ensuring the reliability of the reported bandwidth characterization.

The OE response within DC to 110 GHz are updated with the newly calibrated data in the revised manuscript.

(Main part, Page 3 line 88) In terms of OE bandwidth, the device exhibits a relatively flat frequency response from DC to 170 GHz and achieves a 3 dB bandwidth exceeding 250 GHz (Fig.2f).

Fig.2f OE bandwidth of the modified UTC-PD.

Reference:

- [1] Lischke, S. et al. Ultra-fast germanium photodiode with 3-dB bandwidth of 265 GHz. *Nature Photonics*, 15(12), 925-931 (2021).
- [2] Rouvalis, E. et al. 170 GHz uni-traveling carrier photodiodes for InP-based photonic integrated circuits. *Optics express*, 20(18), 20090-20095 (2012).
- [3] Pozar, D. M. *Microwave engineering*. Fourth Editions, University of Massachusetts at Amherst, John Wiley & Sons (2012).

Comment 5: “What’s the UTC PD responsivity? What are the bandwidth, saturation power, dark current and linear output IP3 of UTC-PD at high-temperature, say 100 °C?”

Our reply: We thank the reviewer for the comments. The external responsivity of the UTC-PD at 1550 nm is 0.24 A/W, as already listed in Table.II of the Methods section.

We also conduct a series of new experiments to characterize the PD’s temperature-dependent performance (25–100 °C), covering responsivity, 3 dB bandwidth, output saturation, dark current, and linearity (OIP3). As shown in Figure.R6, the 3 dB bandwidth gradually decreases from 254 GHz at 25 °C to 196 GHz at 100 °C, while maintaining a smooth frequency-response profile over the entire band. The corresponding saturation RF output power measured at 140 GHz, 180 GHz, and 220 GHz (Figure.R7) follows a consistent trend with the frequency-response degradation. From 25 °C to 100 °C, the saturation RF output power drops from 1.26 dBm to -0.44 dBm for 140 GHz, from -1.40 dBm to -3.42 dBm for 180 GHz and from -3.12 dBm to -6.84 dBm for 220 GHz. The dark current increases from 0.02 nA to 1.61 nA at -1.2 V, remaining comparable to that of typical InP PDs at room temperature (Figure.R8). The measured OIP3 at 20 GHz and 40 GHz shows negligible degradation with temperature, indicating that linearity is unaffected (Figure.R9). A detailed summary of measured parameters is provided in Table.R1. **These results confirm that the operational window of our modified UTC-PD can be extended up to 100 °C with minor temperature-induced variations, demonstrate robust high-speed and linear operation under cruel environment, and make a critical attribute for uncooled or high-temperature photonic-THz applications.**

Figure.R6 Frequency response of the UTC-PD measured at (a) 25 °C, (b) 50 °C, (c) 75 °C, and (d) 100 °C.

Figure.R7 Measured RF output power versus photocurrent at (a) 25 °C, (b) 50 °C, (c) 75 °C, and (d) 100 °C under a bias of 2.2 V. The solid gray line represents the ideal linear relation.

Figure.R8 Measured dark current–voltage curves under different temperatures.

Figure.R9 OIP3 of the UTC-PD measured at 20 GHz and 40 GHz under different temperatures.

Table.R1 Temperature-dependent performance (25–100 °C) of the modified UTC-PD.

Temperature (°C)	25	50	75	100
------------------	----	----	----	-----

Responsivity (A/W)	0.241	0.218	0.206	0.199
Saturated RF Output Power @ 140 GHz (dBm)	1.26	0.65	-0.09	-0.43
Saturated RF Output Power @ 180 GHz (dBm)	-1.4	-1.32	-2.17	-3.42
Saturated RF Output Power @ 220 GHz (dBm)	-3.11	-3.88	-4.83	-6.84
Dark current @ -1.2 V (nA)	0.02	0.07	0.3	1.61
OIP3 @ 20 GHz (dBm)	10.52	11.08	11.33	11.12
OIP3 @ 40 GHz (dBm)	7.95	6.9	6.85	6.77

To comprehensively present the temperature-dependent performance of our modified UTC-PD, we add a new section in the supplementary information:

(Supplementary Note II: Temperature-dependent characterization of the UTC-PD) Temperature-dependent performance is critical for practical applications of PDs, as it directly affects PDs' reliability and robustness under varying environmental conditions. We conduct a series of experiments to characterize the PD's temperature-dependent performance (25–100 °C), covering responsivity, 3 dB bandwidth, output saturation, dark current, and linearity (OIP3). As shown in Figure.S7, the 3 dB bandwidth gradually decreases from 254 GHz at 25 °C to 196 GHz at 100 °C, while maintaining a smooth frequency-response profile over the entire band. The corresponding saturation RF output power measured at 140 GHz, 180 GHz, and 220 GHz (Figure.S8) follows a consistent trend with the frequency-response degradation. From 25 °C to 100 °C, the saturation RF output power drops from 1.26 dBm to -0.44 dBm for 140 GHz, from -1.40 dBm to -3.42 dBm for 180 GHz and from -3.12 dBm to -6.84 dBm for 220 GHz. The dark current increases from 0.02 nA to 1.61 nA at -1.2 V, remaining comparable to that of typical InP PDs at room temperature (Figure.S9). The measured OIP3 at 20 GHz and 40 GHz shows negligible degradation with temperature, indicating that linearity is unaffected (Figure.S10). A detailed summary of measured parameters is provided in Table.S1. These results confirm that the operational window of our modified UTC-PD can be extended up to 100 °C with minor temperature-induced variations, demonstrate robust high-speed and linear operation under cruel environment, and make a critical attribute for uncooled or high-temperature photonic-THz applications.

Figure.S7 Frequency response of the UTC-PD measured at (a) 25 °C, (b) 50 °C, (c) 75 °C, and (d) 100 °C.

Figure.S8 Measured RF output power versus photocurrent at (a) 25 °C, (b) 50 °C, (c) 75 °C, and (d) 100 °C under a bias of 2.2 V. The solid purple line represents the ideal linear relation.

Figure.S9 Measured dark current–voltage curves under different temperatures.

Figure.S10 OIP3 of the UTC-PD measured at 20 GHz and 40 GHz under different temperatures.

Table.S1 Temperature-dependent performance (25–100 °C) of the modified UTC-PD.

Temperature (°C)	25	50	75	100
Responsivity (A/W)	0.241	0.218	0.206	0.199
Saturated RF Output Power @ 140 GHz (dBm)	1.26	0.65	-0.09	-0.44
Saturated RF Output Power @ 180 GHz (dBm)	-1.4	-1.32	-2.17	-3.42
Saturated RF Output Power @ 220 GHz (dBm)	-3.12	-3.88	-4.83	-6.84
Dark current @ -1.2 V (nA)	0.02	0.07	0.3	1.61
OIP3 @ 20 GHz (dBm)	10.52	11.08	11.33	11.12
OIP3 @ 40 GHz (dBm)	7.95	6.9	6.85	6.77

Comment 6: “Were any averaging and filtering used in eye diagrams of Fig. 2(h)?”

Our reply: We thank the reviewer for pointing out this concern. In the eye diagrams tests, we use different hardware bandwidth settings of the DSO. As stated in the operation manual, the DSO's inherent noise floor and the out-of-band noise increase progressively with the expansion of supported bandwidth. **For different data rates (i.e., different signal bandwidths), we try to select the optimal bandwidth configuration, thereby ensuring signal integrity while effectively suppressing the noise.**

At lower data transmission rates, a reduced hardware bandwidth of the DSO is selected, resulting in lower inherent noise floor of the DSO. Clear eye opening can be observed without DSP equalization. When the transmission rate approaches the test equipment's limit (210 Gbaud NRZ & 192 Gbaud PAM-4), the maximum bandwidth of the DSO is activated. **Under this condition, the DSO's inherent noise floor becomes so high that it overwhelms the measured signal. To mitigate this, we employ 4 times averaging to suppress the DSO's noise, thereby more accurately revealing the intrinsic bandwidth performance of our modulator. It is worth noting that this method can only compensate for the signal-to-noise ratio (SNR) but cannot eliminate the signal distortion caused by bandwidth limitation.** No bandwidth compensation DSPs are utilized in the experiment. After averaging, the eye diagrams become clear without any signal distortion, exhibiting the ultra-wideband performance of our modulator. By increasing the modulator length to enhance modulation efficiency, or replacing the EDFA amplification with high-power DFB lasers, fully DSP-free transmission exceeding 200 Gbaud can be expected in the future.

In order to avoid potential misunderstanding, we have revised the original text as follows:

(Main part, Page 3 line 108) ...different symbol rates up to 210 Gbaud NRZ and 196 Gbaud PAM-4 (Supplementary Note 2), **representing the highest speed without bandwidth compensation DSP to date.**

(Methods, Details of data transmission experiments) We apply **no bandwidth compensation DSP** and take screenshots of the eye diagrams to show the intrinsic UWB features of our TFLN modulator.

(Methods, Comparison of the UWB integrated photonics system) Table.III shows the comparison of the integrated IMDD communications. Our approach demonstrates the highest transmission rate **without any bandwidth compensation DSP.**

Comment 7: "In Fig. 2(i), BER is shown with complex-biGRU post-processing. Any EDFA noise loading? Please discuss equalizer tap counts/complexity used to reach the HD-FEC threshold."

Our reply: Thanks for the comments. For both wired and wireless transmission experiments, signal lights are first amplified by an EDFA before sending into PDs. Therefore, the pre-equalization signal incorporates the noise effect of EDFA. Besides EDFA noise, noise and signal distortions from RF devices, free space and other components in the transmission system are also encompassed and collectively influence the overall transmission performance. The proposed complex-biGRU algorithm demonstrates extraordinary capability in handling these generalized nonlinear problems and thus results in record-level BER performance. To comprehensively demonstrate the performance of our proposed complex-biGRU algorithm, we compare it with other commonly used algorithms in terms of computational complexity, power efficiency, latency, and equalization capability, as presented below and in **Comment 8 & 10.**

For complexity analysis, considering that both Volterra nonlinear equalizers (VNLEs) and neural networks (NNs) are inherently formulated in terms of matrix or tensor operations, where the dominant

computational burden arises from multiplications between the weight matrices and the input data, we adopt the multiply–accumulate operation (MAC) as a representative metric for complexity comparison. A single MAC corresponds to one multiplication followed by one accumulation within the matrix multiplication process.

The computational complexity of N-th order VNE can be expressed as:

$$C_{VNLE} = \sum_{n=1}^N \frac{(L_n - 1 + n)!}{(n-1)!(L_n - 1)!} \quad (R8)$$

where L_n refers to the n-th order memory length.

The computational complexity of a two-hidden-layer deep neural network (DNN) can be expressed as:

$$C_{DNN} = n_0 n_1 + n_1 n_2 + n_2 n_3 \quad (R9)$$

where n_0 refers to the size of input layer, n_3 represents the size of output layer, and n_1/n_2 denote the number of neurons in the first and second hidden layers respectively.

The main computation complexity of one GRU layer can be described as:

$$C_{GRU} = 3n_H(n_E + n_H) \quad (R10)$$

where n_E refers to the input size of the GRU layer, and n_H represents the number of GRU units that used in the hidden layer. Based on this equation, the complexity of bi-GRU and complex-biGRU layer can be calculated as:

$$C_{bi-GRU} = 2 \times 3n_H(n_E + n_H) \quad (R11)$$

$$C_{complex-biGRU} = 2 \times 2 \times 3n_H(n_E + n_H) \quad (R12)$$

The computational complexity of different algorithms is summarized in Table.R2. To provide a more intuitive comparison of complexities, we calculated the floating-point operations (FLOPs) per symbol for the three algorithms involved in the comparative analysis. For the VNLE algorithm, we employ a second-order Volterra model (i.e., N=2), and the order memory length is set to 80. The number of MACs can be calculated by:

$$C_{VNLE} = \sum_{n=1}^N \frac{(L_n - 1 + n)!}{(n-1)!(L_n - 1)!} = \sum_{n=1}^2 \frac{(80 - 1 + n)!}{(n-1)!(80 - 1)!} = 6560$$

So, the required FLOPs per symbol of VNLE are $6560 \times 2 = 13120$. For the DNN algorithm, the size of input layer, first/second hidden layer, and output layers are 30, 40, 40 and 1 respectively. The number of MACs can be then calculated as:

$$C_{DNN} = n_0 n_1 + n_1 n_2 + n_2 n_3 = 30 \times 40 + 40 \times 40 + 40 \times 1 = 2840$$

So, the required FLOPs per symbol of DNN are $2840 \times 2 = 5680$. For the complex-biGRU algorithm, the number of GRU units is set to 200. In wired transmission scenario, only one branch of the network is activated and the input size of the GRU layer is $2^{15} - 1$. While in the wireless transmission scenario, the input size of the GRU layer is 10^5 . The number of MACs can be then calculated as:

$$C_{complex-biGRU,wired} = 2 \times 3n_H(n_E + n_H) = 2 \times 3 \times 200 \times (200 + 2^{15} - 1) = 3.956 \times 10^7$$

$$C_{complex-biGRU,wireless} = 2 \times 2 \times 3n_H(n_E + n_H) = 2 \times 2 \times 3 \times 200 \times (200 + 10^5) = 2.405 \times 10^8$$

So, the required FLOPs per symbol of complex-biGRU are $(2 \times 3.956 \times 10^7) / (2^{15} - 1) \approx 2414$ for wired scenario and $(2 \times 2.405 \times 10^8) / 10^5 \approx 4809$ for wireless scenario. **We can see that the number of FLOPs of the complex-biGRU algorithm is at a comparable order of magnitude to that of other neural network algorithms or the VNLE algorithm, indicating that the complex-biGRU algorithm does not increase computational complexity.**

Table.R2 Comparison of computational complexity among different algorithms.

Algorithm	MAC
VNLE	$\sum_{n=1}^N \frac{(L_n - 1 + n)!}{(n-1)! (L_n - 1)!}$
DNN	$n_0 n_1 + n_1 n_2 + n_2 n_3$
GRU	$3n_H(n_E + n_H)$
bi-GRU	$2 \times 3n_H(n_E + n_H)$
complex-biGRU	$2 \times 2 \times 3n_H(n_E + n_H)$

In order to clarify the complexity settings used to reach the HD-FEC threshold, we further provide a detailed evaluation of the complex-biGRU algorithm's performance under different network complexities. Specifically, we investigate complex-biGRU equalization with different number of hidden GRU units for 50 Gbaud 16-QAM and 70 Gbaud 16-QAM transmission, as depicted in Figure.R10. As the number of GRU units increase, the algorithm's channel equalization capability gradually improves, with the complete channel response being progressively modeled, leading to a reduction in BER. However, when the number of GRU units reached a certain threshold (i.e., 200 GRU units in this case), the BER ceased to decrease further, exhibiting a saturation region. Continued increase in GRU units will lead to overfitting and higher power consumption. Thus, our strategy is to employ the minimum GRU units of the saturation region at each speed to balance performance and efficiency.

Figure.R10 BER curves of different GRU numbers in the hidden layer.

The corresponding results and analysis are added to the revised manuscript and supplementary information:

(Supplementary Note V) ...demonstrating constellation points that conform to a 2D Gaussian distribution.

To comprehensively demonstrate the performance of our proposed complex-biGRU algorithm, we compare it with other commonly used algorithms in terms of computational complexity, power efficiency, latency, and equalization capability.

For complexity analysis, we compare the complex-biGRU algorithm with two typical nonlinear equalization methods - Volterra nonlinear equalizers (VNLE) and neural networks (NN). Considering that both VNLEs and NNs are inherently formulated in terms of matrix or tensor operations, where the dominant computational burden arises from multiplications between the weight matrices and the input data, we adopt the multiply-accumulate operation (MAC) as a representative metric for complexity comparison. A single MAC corresponds to one multiplication followed by one accumulation within the matrix multiplication process. Table.S2 summarizes the computational complexity of several algorithms. For VNLE, L_n refers to the n-th order memory length. For two-layer deep neural networks (DNN), n_0

refers to the size of input layer, n_3 represents the size of output layer, and n_1/n_2 denotes the number of neurons in the first and second hidden layers respectively. For GRU based algorithms, n_E refers to the input size of the GRU layer, and n_H represents the number of GRU units that used in the hidden layer. To provide a more intuitive comparison of complexities, we calculated the floating-point operations (FLOPs) per symbol for the three algorithms involved in the comparative analysis. For the VNLE algorithm, we employ a second-order Volterra model (i.e., $N=2$), and the order memory length is set to 80. The number of MACs can be calculated by:

$$C_{VNLE} = \sum_{n=1}^N \frac{(L_n-1+n)!}{(n-1)!(L_n-1)!} = \sum_{n=1}^2 \frac{(80-1+n)!}{(n-1)!(80-1)!} = 6560$$

So, the required FLOPs per symbol of VNLE are $6560 \times 2 = 13120$. For the DNN algorithm, the size of input layer, first/second hidden layer, and output layers are 30, 40, 40 and 1 respectively. The number of MACs can be then calculated as:

$$C_{DNN} = n_0n_1 + n_1n_2 + n_2n_3 = 30 \times 40 + 40 \times 40 + 40 \times 1 = 2840$$

So, the required FLOPs per symbol of DNN are $2840 \times 2 = 5680$. For the complex-biGRU algorithm, the number of GRU units is set to 200. In wired transmission scenario, only one branch of the network is activated and the input size of the GRU layer is $2^{15}-1$. While in the wireless transmission scenario, the input size of the GRU layer is 10^5 . The number of MACs can be then calculated as:

$$C_{complex-biGRU,wired} = 2 \times 3n_H(n_E + n_H) = 2 \times 3 \times 200 \times (200 + 2^{15} - 1) = 3.956 \times 10^7$$

$$C_{complex-biGRU,wireless} = 2 \times 2 \times 3n_H(n_E + n_H) = 2 \times 2 \times 3 \times 200 \times (200 + 10^5) = 2.405 \times 10^8$$

So, the required FLOPs per symbol of complex-biGRU are $(2 \times 3.956 \times 10^7) / (2^{15} - 1) \approx 2414$ for wired scenario and $(2 \times 2.405 \times 10^8) / 10^5 \approx 4809$ for wireless scenario. We can see that the number of FLOPs of the complex-biGRU algorithm is at a comparable order of magnitude to that of other neural network algorithms or the VNLE algorithm, indicating that the complex-biGRU algorithm does not increase computational complexity.

Table.S2 Comparison of computational complexity among different algorithms.

Algorithm	MAC
VNLE	$\sum_{n=1}^N \frac{(L_n - 1 + n)!}{(n - 1)!(L_n - 1)!}$
two-layer DNN	$n_0n_1 + n_1n_2 + n_2n_3$
GRU	$3n_H(n_E + n_H)$
bi-GRU	$2 \times 3n_H(n_E + n_H)$
complex-biGRU	$2 \times 2 \times 3n_H(n_E + n_H)$

We also evaluate the complex-biGRU algorithm's performance under different network complexities. Specifically, we investigate complex-biGRU equalization with different number of hidden GRU units for 50 Gbaud 16-QAM and 70 Gbaud 16-QAM transmission, as depicted in Figure.S29. As the number of GRU units increase, the algorithm's channel equalization capability gradually improves, with the complete channel response being progressively modeled, leading to a reduction in BER. However, when the number of GRU units reached a certain threshold (i.e., 200 GRU units in this case), the BER ceased to decrease further, exhibiting a saturation region. Continued increase in GRU units will lead to overfitting and higher power consumption. Thus, our strategy is to employ the minimum GRU units of the saturation region at each speed to balance performance and efficiency.

Figure.S29 BER curves of different GRU numbers in the hidden layer.

Comment 8: “What would be the raw BER without the complex-biGRU? How much latency and power it would incur to run the complex-biGRU?”

Our reply: We appreciate the reviewer’s comments about power consumption and latency, which are important for practical application scenarios. The comparison of raw BER and BER after complex-biGRU under different modulation formats is shown in Figure.R11. The equalized BER improves by several orders of magnitude compared with the raw BER, which demonstrates the extraordinary equalization capability of the proposed complex-biGRU algorithm. Other equalization algorithms can also be used to reduce the BER below the decision threshold, with the comparison shown in Comment 10.

Figure.R11 Comparison of raw BER and BER after complex-biGRU for (a) NRZ, (b) PAM4, (c) 16QAM, and (d) 32QAM.

To get the power consumption data, we use HWINFO [<https://www.hwinfo.com/>] to monitor the hardware status when running different equalization algorithms. A computer equipped with an Intel Core i9-13900HX processor, 32 GB of 5600 MHz memory, and an Nvidia RTX 4060 GPU is employed in the experiment. During testing with the complex-biGRU, the measured power consumption is approximately 38.152 W for NRZ signal and 42.572 W for PAM-4 signal in the wired scenario, whereas it increases to about 50.64 W for 16-QAM signal and 50.78 W for 32-QAM signal in the wireless scenario due to the more complicated channel environment. Furthermore, we introduce the equalizer power consumption per bit (EPCpB) as a comparison metric for different speeds, which is defined as:

$$\text{EPCpB} = \text{Power Consumption (W)} / \text{Data Rate (Gbps)}$$

A smaller EPCpB value indicates a lower power consumption per bit. We also evaluate and compare the power consumption of traditional nonlinear algorithms and other AI equalizers as listed in Table.R3. The

maximum transmission rate below SD-FEC/HD-FEC threshold is selected as the reference speed for comparison. **It can be seen that in simple channel environments, the complex-biGRU algorithm achieves higher transmission rates with comparable power consumption. For the complex channel circumstances such as wireless communication or ultra-high-speed communication, the complex-biGRU algorithm does not significantly increase the EPCpB and even reduce it due to a much larger data rate. In other words, it is worthwhile to use higher equalizer power consumption in exchange for an improvement in transmission speed based on AI equalizers.**

Table.R3 Comparison of power consumption for different algorithms under different scenarios.

Equalizer	Power (W)	Signal format	Speed (Gbaud)	BER threshold	EPCpB (nJ/bit)
VNLE	~40.09	NRZ	240	HD-FEC	0.1670
VNLE	~42.25	PAM-4	168	SD-FEC	0.1257
DNN	~12.79	NRZ	250	HD-FEC	0.0512
DNN	~17.15	PAM-4	154	HD-FEC	0.0557
complex-biGRU	~38.15	NRZ	256	HD-FEC	0.1490
complex-biGRU	~42.57	PAM-4	256	HD-FEC	0.0831
VNLE	~71.03	16-QAM	80	SD-FEC	0.2220
VNLE	~71.23	32-QAM	80	SD-FEC	0.1781
complex-biGRU	~50.64	16-QAM	70	HD-FEC	0.1808
complex-biGRU	~50.78	32-QAM	60	HD-FEC	0.1693

For latency analysis, the latency introduced by the equalizer depends on several factors, primarily including inference latency and data transfer latency.

1) Inference latency

Inference latency refers to the time required for an algorithm to process an input signal and generate the corresponding output. The inference latency depends on: a) Number of operations: each layer in a network performs a certain number of mathematical operations. The total number of operations can be calculated based on the layer sizes, layer types, and input/output sizes; b) Hardware throughput: the type of processing hardware (e.g., CPU, GPU or FPGA) determines how quickly the operations can be performed. Different hardware platforms have different processing speeds measured in floating-point operations per second (FLOPS). The inference latency can be calculated by:

$$L_{inference} = \text{Total Operations} / \text{Hardware Throughput}$$

where total operations refer to the total number of FLOPs required to process one input, and hardware throughput refers to the number of FLOPs the hardware can handle per second.

2) Data transfer latency

Data transfer latency refers to the time required to move input data from memory to the processing unit and the output from the processing unit back to memory or to the next processing stage. This depends on: a) Memory bandwidth: the speed at which data can be read from and written to memory; b) Data size: the size of the data needs to be processed. Data transfer latency can be estimated as:

$$L_{transfer} = \text{Data Size} / \text{Memory Bandwidth}$$

The total latency L_{total} can be expressed as:

$$L_{total} = L_{inference} + L_{transfer}$$

In our experiment, the received signal is sampled by a real-time oscilloscope and stored for offline digital signal processing and BER calculation. Data processing is carried out using the same computer setup as we mentioned above. The hardware throughput, as specified in the device specifications, is 844.8 GfLOPs for the CPU and 14.56 TFLOPs for the GPU respectively.

For the VNLE, the required FLOPs per symbol are 13120. When processed by the CPU, the latency equals to:

$$L_{inference_VNLE_CPU} = 13120 / 844.8 \times 10^9 = 15.530 \text{ ns}$$

In the case of GPU,

$$L_{inference_VNLE_GPU} = 13120 / 14.56 \times 10^{12} = 0.901 \text{ ns}$$

For the DNN algorithm, the inference process requires 5680 FLOPs. When processed by CPU, the latency equals to:

$$L_{inference_DNN_CPU} = 5680 / 844.8 \times 10^9 = 6.724 \text{ ns}$$

In the case of GPU,

$$L_{inference_DNN_GPU} = 5680 / 14.56 \times 10^{12} = 0.390 \text{ ns}$$

For complex-biGRU, in wired transmission scenario, the required FLOPs per symbol are 2414. When processed by the CPU, the latency equals to:

$$L_{inference_wired_complex-biGRU_CPU} = 2414 / 844.8 \times 10^9 = 2.858 \text{ ns}$$

In the case of GPU,

$$L_{inference_wired_complex-biGRU_GPU} = 2414 / 14.56 \times 10^{12} = 0.166 \text{ ns}$$

In the wireless transmission scenario, the required FLOPs of per symbol are 4809. When processed by the CPU, the latency equals to:

$$L_{inference_wireless_complex-biGRU_CPU} = 4809 / 844.8 \times 10^9 = 5.693 \text{ ns}$$

In the case of GPU,

$$L_{inference_wireless_complex-biGRU_GPU} = 4809 / 14.56 \times 10^{12} = 0.330 \text{ ns}$$

For the data transfer latency, the memory bandwidth can be calculated by:

Memory Bandwidth = Memory Frequency \times Data Transferred per Cycle \times Number of Memory Channels

which equals to $5.6 \text{ G} \times 8 \text{ B} \times 2 = 89.6 \text{ GB/s}$ in our system. For wired transmission scenario, the data size is 0.294 MB. For wireless transmission scenario, the data size is 0.808 MB. We then calculate the average data transfer latency for each symbol using CPU as:

$$L_{wired_transfer, DDR5} = (0.294 \text{ MB} / 89.6 \text{ GB}) / (2^{15}-1) = 0.100 \text{ ns}$$

$$L_{wireless_transfer, DDR5} = (0.808 \text{ MB} / 89.6 \text{ GB}) / 10^5 = 0.090 \text{ ns}$$

Additionally, when GPU computing is employed, the data will be transferred through GPU memory

rather than standard DDR5. Since GPU memory typically provides much higher bandwidth compared to general-purpose system memory, the data transfer latency between the GPU and its memory can be significantly reduced. The GPU memory bandwidth of our computer is 288 GB/s, so the average data transfer latency for each symbol can be calculated as:

$$L_{\text{wired_transfer, GDDR}} = (0.294 \text{ MB} / 288 \text{ GB}) / (2^{15}-1) = 0.031 \text{ ns}$$

$$L_{\text{wireless_transfer, GDDR}} = (0.808 \text{ MB} / 288\text{GB}) / 10^5 = 0.028 \text{ ns}$$

We summarize the latency of each equalization algorithms under different transmission scenarios in Table.R4. We can see that the total latency of different algorithms is comparable and using GPU can significantly reduce the total latency. It's worth noting that the latency here only includes the time that equalizer takes to process a single symbol. The overall system latency also includes the time consumed by resampling, clock synchronization, signal decision-making and other signal processing procedures. Currently, the latency is measured and calculated based on a generalized platform, which can be further reduced through precision algorithm optimization and employing ASIC based hardware.

Table.R4 Comparison of equalizer latency for different algorithms under different scenarios.

Scenario	Equalizer	Inference latency	Data transfer latency	Total latency
Wired	VNLE	CPU: 15.530 ns	DDR: 0.100 ns	CPU: 15.630 ns
		GPU: 0.901 ns	GDDR: 0.031 ns	GPU: 0.932 ns
	DNN	CPU: 6.724 ns	DDR: 0.100 ns	CPU: 6.824 ns
		GPU: 0.390 ns	GDDR: 0.031 ns	GPU: 0.421 ns
	complex-biGRU	CPU: 2.858 ns	DDR: 0.100 ns	CPU: 2.958 ns
		GPU: 0.166 ns	GDDR: 0.031 ns	GPU: 0.197 ns
Wireless	VNLE	CPU: 15.530 ns	DDR: 0.090 ns	CPU: 15.620 ns
		GPU: 0.901 ns	GDDR: 0.028 ns	GPU: 0.929 ns
	complex-biGRU	CPU: 5.693 ns	DDR: 0.090 ns	CPU: 5.783 ns
		GPU: 0.330 ns	GDDR: 0.028 ns	GPU: 0.358 ns

To further clarify the inference latency and power efficiency of the complex-biGRU algorithm, we have added the following descriptions in the supplementary information:

(Supplementary Note V) For power consumption analysis, we introduce the equalizer power consumption per bit (EPCpB) as a comparison metric for different speeds, which is defined as:

$$\text{EPCpB} = \text{Power Consumption (W)} / \text{Data Rate (Gbps)}$$

A smaller EPCpB value indicates a lower power consumption per bit. We evaluate and compare the power consumption of traditional nonlinear algorithms and other AI equalizers as listed in Table.S3. The maximum transmission rate below SD-FEC/HD-FEC threshold is selected as the reference speed for comparison. It can be seen that in simple channel environments, the complex-biGRU algorithm achieves higher transmission rates with comparable power consumption. For the complex channel circumstances such as wireless communication or ultra-high-speed communication, the complex-biGRU algorithm does not significantly increase the EPCpB and even reduce it due to a much larger data rate. In other words, it is worthwhile to use higher equalizer power consumption in exchange for an improvement in

transmission speed based on AI equalizers.

Table.S3 Comparison of power consumption for different algorithms under different scenarios.

Equalizer	Power (W)	Signal format	Speed (Gbaud)	BER threshold	EPCpB (nJ/bit)
VNLE	~40.09	NRZ	240	HD-FEC	0.1670
VNLE	~42.25	PAM-4	168	SD-FEC	0.1257
DNN	~12.79	NRZ	250	HD-FEC	0.0512
DNN	~17.15	PAM-4	154	HD-FEC	0.0557
complex-biGRU	~38.15	NRZ	256	HD-FEC	0.1490
complex-biGRU	~42.57	PAM-4	256	HD-FEC	0.0831
VNLE	~71.03	16-QAM	80	SD-FEC	0.2220
VNLE	~71.23	32-QAM	80	SD-FEC	0.1781
complex-biGRU	~50.64	16-QAM	70	HD-FEC	0.1808
complex-biGRU	~50.78	32-QAM	60	HD-FEC	0.1693

We also calculate and compare the latency of different algorithms. The latency introduced by the equalizer depends on several factors, primarily including inference latency and data transfer latency. Inference latency refers to the time required for an algorithm to process an input signal and generate the corresponding output, which can be calculated by:

$$L_{\text{inference}} = \text{Total Operations} / \text{Hardware Throughput}$$

Data transfer latency refers to the time required to move input data from memory to the processing unit and the output from the processing unit back to memory or to the next processing stage, which can be estimated as:

$$L_{\text{transfer}} = \text{Data Size} / \text{Memory Bandwidth}$$

Therefore, the total delay can be calculated as the sum of the inference latency and data transfer latency.

$$L_{\text{total}} = L_{\text{inference}} + L_{\text{transfer}}$$

Based on the equations above, we calculate and summarize the average latency per symbol of each equalization algorithm under different transmission scenarios using different hardware in Table.S4. We can see that the total latency of different algorithms is comparable and using GPU can significantly reduce the total latency. It's worth noting that the latency here only includes the time that equalizer takes to process a single symbol. The overall system latency also includes the time consumed by resampling, clock synchronization, signal decision-making and other signal processing procedures. Currently, the latency is measured and calculated based on a generalized platform, which can be further reduced through precision algorithm optimization and employing ASIC based hardware.

Table.S4 Comparison of equalizer latency for different algorithms under different scenarios.

Scenario	Equalizer	Inference latency	Data transfer latency	Total latency
Wired	VNLE	CPU: 15.530 ns	DDR: 0.100 ns	CPU: 15.630 ns
		GPU: 0.901 ns	GDDR: 0.031 ns	GPU: 0.932 ns

	DNN	CPU: 6.724 ns	DDR: 0.100 ns	CPU: 6.824 ns
		GPU: 0.390 ns	GDDR: 0.031 ns	GPU: 0.421 ns
	complex-biGRU	CPU: 2.858 ns	DDR: 0.100 ns	CPU: 2.958 ns
		GPU: 0.166 ns	GDDR: 0.031 ns	GPU: 0.197 ns
Wireless	VNLE	CPU: 15.530 ns	DDR: 0.090 ns	CPU: 15.620 ns
		GPU: 0.901 ns	GDDR: 0.028 ns	GPU: 0.929 ns
	complex-biGRU	CPU: 5.693 ns	DDR: 0.090 ns	CPU: 5.783 ns
		GPU: 0.330 ns	GDDR: 0.028 ns	GPU: 0.358 ns

Comment 9: “I wouldn’t say 6.2 V of $V_{\pi,RF}$ low.”

Our reply: We apologize for any confusion caused to the reviewer. Our intention of using “low” in the sentence is to address that our modulator achieves “lower” $V_{\pi,RF}$ compared with other ultra-wideband modulator structures as illustrated in Table.R5. Yet, we acknowledge that the absolute value of 6.2 V is not low enough.

Table.R5 Comparison of the $V_{\pi,RF}$ of different modulator structure.

Structure	TFLT [1]	Plasmonic [2]	TFLN [3]	TFLN [4]	This work
$V_{\pi,RF}$	6.78 V @ 110 GHz	23.4V @ 200 GHz	12.5 V @ 200 GHz	7.5V @ 200 GHz	6.2V @ 200 GHz

To prevent any confusion, we have modified the description as follows:

(Main part, Page 3 line 57) Benefiting from the excellent EO bandwidth, radio frequency (RF) half-wave voltage ($V_{\pi,RF}$) of 6.2 V at 200 GHz can be calculated (Methods).

Reference:

- [1] Wang, C. et al. Ultrabroadband thin-film lithium tantalate modulator for high-speed communications. *Optica* 11, 1614–1620 (2024).
- [2] Horst, Y. et al. Ultra-wideband MHz to THz plasmonic EO modulator. *Optica* 12, 325–328 (2025).
- [3] Mercante, A. J. et al. Thin film lithium niobate electro-optic modulator with terahertz operating bandwidth. *Optics express* 26(11), 14810-14816 (2018).
- [4] Zhang, Y. et al. Systematic investigation of millimeter-wave optic modulation performance in thin-film lithium niobate. *Photonics Research* 10, 2380–2387 (2022).

Comment 10: “For the complex-biGRU, please specify model size, training data volume, compute time, and inference latency; compare against practical real-time budgets.”

Our reply: Thanks for the comment. We summarize the training metrics in Table.R6. A more detailed explanation and comparison, including complexity, power efficiency and latency can be found in our response to Comment 7-8.

Table.R6 Training metrics structure in wired and wireless scenarios.

	Wired	Wireless
Model size	$2 \times 2 \times 3 n_H (n_E + n_H)$	$2 \times 2 \times 2 \times 3 n_H (n_E + n_H)$
Training data volume	30K	100K

Compute time	30min	2h10min
Inference latency	0.02036 ms	0.04004 ms

To further measure the equalization ability of complex-biGRU and other traditional algorithms, we employ various DSP algorithms to equalize the same set of data collected under different scenarios and rates. The tap weights of DFE are updated by least mean squares (LMS) algorithm and third order VNLE is employed. In wired transmission, the performance of DFE, VNLE, DNN and complex-biGRU for high-speed NRZ and PAM-4 signals is evaluated, and the results are shown in Figure.R12. The results demonstrate that, compared to DFE and VNLE, complex-biGRU can achieve significant performance improvements for NRZ and PAM-4 signals, showing an improvement of one or two orders of magnitude in BER. For wireless transmission scenarios, the comparative performance of VNLE and complex-biGRU for high-speed 16-QAM and 32-QAM signals is evaluated, with the results presented in Figure.R13. The results reveal that complex-biGRU also yields remarkable enhancements in BER over VNLE, especially for high-speed signal with large signal bandwidth.

Figure.R12 BER performance of DFE, VNLE, DNN and complex-biGRU in wired scenarios for (a) NRZ and (b) PAM-4 signal transmission.

Figure.R13 BER performance of VNLE and complex-biGRU in wireless scenarios for (a) 16-QAM and (b) 32-QAM signal transmission.

Despite demonstrating powerful equalization capabilities, the complex-biGRU algorithm still exhibits some inherent limitations in its current implementation. The prolonged training time of the network hinders its rapid redeployment and adaptation across diverse channels. Substantial inference latency and power consumption increase the operational costs in practical scenarios, posing challenges to meet the requirements of real-time applications. Nevertheless, a number of approaches have been devised to mitigate these issues. Transfer learning can be employed to enhance learning efficiency during cross-channel redeployment, thereby avoiding redundant computation [1]. We can also perform systematical parameters tuning of the algorithm to ensure the compatibility with mature computing architectures (e.g., CUDA, PyTorch), thereby reducing both training time and inference latency [2]. Furthermore, application-specific integrated circuit (ASIC) chips tailored for the GRU algorithm can be designed to enable optimized inference performance with lower latency and improved energy efficiency [3]. Overall, the complex-biGRU algorithm consistently outperforms other equalizers in dealing with nonlinear distortions for both wired and wireless scenarios and will play an important role in the next-generation

ultra-high-speed fiber-wireless communication.

To emphasize the superior equalization performance over other DSP algorithms, we have added BER comparisons for both fiber and wireless scenarios, as outlined below:

(Supplementary Note V) To more intuitively compare the equalization capacities of different algorithms, we employ various DSP algorithms to equalize the same set of data collected under different scenarios and rates. In wired transmission, we evaluate the performance of DFE, VNLE, DNN and complex-biGRU for high-speed NRZ and PAM-4 signals as illustrated in Figure.S30. The results demonstrate that, compared to DFE and VNLE, complex-biGRU can achieve significant performance improvements for NRZ and PAM-4 signals, showing an improvement of one or two orders of magnitude in BER. For wireless transmission scenarios, the comparative performance of VNLE and complex-biGRU for high-speed 16-QAM and 32-QAM signals is evaluated, with the results presented in Figure.S31. The results reveal that complex-biGRU also yields remarkable enhancements in BER over VNLE, especially for high-speed signal with large signal bandwidth.

Figure.S30 BER performance of DFE, VNLE, DNN and complex-biGRU in wired scenarios for (a) NRZ and (b) PAM-4 signal transmission.

Figure.S31 BER performance of VNLE and complex-biGRU in wireless scenarios for (a) 16-QAM and (b) 32-QAM signal transmission.

Reference:

- [1] Zhang, J. et al. Meta-learning assisted source domain optimization for transfer learning based optical fiber nonlinear equalization. *Journal of Lightwave Technology*, 41(5), 1269-1277 (2022).
- [2] Paszke, A. et al. Pytorch: An imperative style, high-performance deep learning library. In *2019 Advances in neural information processing systems*, 32.
- [3] Aguirre, F. et al. Hardware implementation of memristor-based artificial neural networks. *Nature communications*, 15(1), 1974 (2024).

Comment 11: “In the 180 GHz wireless experiments, what RF power reached the transmit horn, and what were the antenna gains, cable losses? Provide a complete equivalent isotropically radiated power and receiver sensitivity budget.”

Our reply: We thank the reviewer for the comments. In the 180 GHz wireless experiments, we use 2 mA photocurrent for THz signal generation, corresponding to -14 dBm of RF output power according to

Fig.2e. Since no electrical cables are employed in the experiment, the power delivers to the antenna can be derived by subtracting the probe loss from the power generated by the PD, which is about -16.9 dBm. The relationship between antenna gains and frequency is demonstrated in Figure.R14, where the gain reaches approximately 26 dBi at 180 GHz. Based on the analysis above and the radiation pattern of the antenna, we can calculate the equivalent isotropically radiated power (EIRP) as illustrated in Figure.R15. We also evaluate the sensitivity of our THz receiver. The free space path loss (FSPL) can be calculated using the following equation:

$$FSPL = 20 \times \lg(d) + 20 \times \lg(f) + 32.44 - G_t - G_r \quad (R13)$$

where d is the propagation distance of signal in meters, f is the signal frequency in GHz, and G_t and G_r are the gains of transmitter and receiver antenna. By subtracting the FSPL from the output power of the PD, the received power at the input of the Rx amplifier can be calculated. The emitted power of the UTC-PD can be precisely regulated by adjusting the incident optical power. Therefore, the BER curve under different received power could be obtained to evaluate the receiver sensitivity budget. We employ 70 Gbaud QPSK transmission to test the receiver sensitivity and the BER results are illustrated in Figure.R16. It can be seen that BER decreases with increasing optical power. When the received power exceeds -23 dBm, the BER performance also deteriorates due to amplifier saturation.

Figure.R14 Antenna gains at different frequencies.

Figure.R15 Equivalent isotropically radiated power at 180 GHz.

Figure.R16 BER curves of different receiving power.

To clarify the sensitivity of our optical THz receiver, we have added the following descriptions in the supplementary information:

(Supplementary Note IV) In addition to the transmission rate, we also characterize several other critical metrics in the wireless system. For the sensitivity of our THz receiver, free space path loss (FSPL) can be calculated using the following equation:

$$FSPL = 20 \times \lg(d) + 20 \times \lg(f) + 32.44 - G_t - G_r \quad (S4)$$

where d is the propagation distance of signal in meters, f is the signal frequency in GHz, and G_t and G_r are the gains of transmitter and receiver antenna. By subtracting the FSPL from the output power of the PD, the received power at the input of the Rx amplifier can be calculated. The emitted power of the UTC-PD can be precisely regulated by adjusting the incident optical power. The BER curve of 70 Gbaud QPSK transmission under different received power is obtained (Figure.S18) for example. It can be seen that BER decreases with increasing received power. When the received power exceeds -23 dBm, the BER performance also deteriorates due to amplifier saturation.

Figure.S18 BER curves of different receiving power.

Comment 12: “How are the 86 channels generated and multiplexed (optically/electrically)? What guard bands, adjacent-channel isolation, and aggregate EVM/BER did you achieve across 138–223 GHz?”

Our reply: Thanks for the comments. In our proof-of-concept demonstration, the 86 channels video signals are generated independently using SFP+ optical modules and aggregated through an optical multiplexer. By adjusting the Tx LO frequency, we allocate each signal to its corresponding carrier frequency and transmit them via a time-multiplexed strategy. Passing through the optical-THz-optical conversion, the composite optical signal is separated into baseband signals using another optical wavelength demultiplexer and sent into Rx optical modules for demodulation.

We use an electrical spectrum analyzer (ESA) to get the spectrum of the video signal as shown in Figure.R17. The signal demonstrates a real-time bandwidth of 100 MHz with over 20 dB SNR, which also indicates an adjacent-channel isolation over 20 dB. With a channel spacing of 1 GHz, the guard interval can be calculated as 1000 MHz - 100 MHz = 900 MHz.

Figure.R17 Spectrum of the video signal.

Since the signals of each channel are independently generated, transmitted and demodulated, there is

no aggregate EVM/BER and only the individual BER per channel is assessed. According to the specification of HDMI 2.1 [1], character (with 18 bits in each character) error rates of 10^{-4} or higher are readily detectable by visual inspection. Our demonstration achieved consecutive and clear live video transmission across all 86 channels with no visible error, indicating that the BER of each channel remained below $10^{-4} / 18 = 5.5 \times 10^{-6}$.

To provide a full view of our multi-channel access system, we have added the above analysis in the supplementary information:

(Supplementary Note IV) Besides characterizing the high-speed wireless transmission system, we also evaluate the performance of the multi-channel video transmission system. We use an electrical spectrum analyzer (ESA) to get the spectrum of the video signal as shown in Figure.S24. The signal demonstrates a real-time bandwidth of 100 MHz with over 20 dB SNR, which also indicates an adjacent-channel isolation over 20 dB. With a channel spacing of 1 GHz, the guard interval can be calculated as $1000 \text{ MHz} - 100 \text{ MHz} = 900 \text{ MHz}$.

Figure.S24 Spectrum of the video signal.

The BER performance of each channel is also assessed. According to the specification of HDMI 2.1 [9], character (with 18 bits in each character) error rates of 10^{-4} or higher are readily detectable by visual inspection. Our demonstration achieves consecutive and clear live video transmission across all 86 channels with no visible error, indicating that the BER of each channel remains below $10^{-4} / 18 = 5.5 \times 10^{-6}$.

Reference:

[1] HDMI Forum. High-Definition Multimedia Interface Specification Version 2.1. (2017).

Comment 13: “Was the video demo truly simultaneous across 86 channels, or time-multiplexed? Please provide aggregate throughput and power.”

Our reply: As we mentioned in Comment 12, the 86-channel video transmission demo is performed using a time-multiplexed scheme. Nevertheless, we have successfully demonstrated the parallel connection ability of our system with two video stream signals modulated onto 155 GHz channel and 172 GHz channel simultaneously, showcasing the feasibility of future multi-channel parallel transmission. Currently, the primary limitation of simultaneous multi-channel video transmission is the coarse filtering bandwidth of FBGs, which allows multiple signals to coexist within the filtering bandwidth, causing severe crosstalk. In the future, we can achieve precise channel selection by utilizing on-chip finely tunable microwave photonic filters with high suppression ratios [1], thereby enabling truly simultaneous multi-channel transmission.

In the video transmission experiment, the computers connect to the Tx/Rx switch via Ethernet ports

under IEEE 802.3ab protocol with 1 Gbps maximum speed. Thus, the maximum aggregate throughput of our multi-channel system is limited to $86 \times 1 \text{ Gbps} = 86 \text{ Gbps}$. We also present the power consumption breakdown of the video transmission system as shown in Figure.R18. At the transmitter side, the switch served for signal routing consumes 43 W. The power consumption of two optical modules is 1.8 W each. The tunable ECL serves as the Tx LO with a power consumption of 50 W. The EDFA amplifies the mixed signals with a power consumption of 24 W. The amplified optical signal is then sent to the modified UTC-PD, consuming only 0.0044 W (2.2 V, 2 mA). At the receiver side, the signal is amplified by a low-noise amplifier (0.054 W) before sent into the TFLN modulator (0.045 W) for THz-to-optical conversion. The high-power laser driving the TFLN modulator is 20 W. The EDFA power consumption at the receiver side is 24 W. Two Rx optical modules consume 1.8 W each. And the power consumption of the Rx switch is 43 W. The overall system consumption is 211.3034 W. It is worth noting that in the current system, two of the major power consumption equipment are tunable ECL and EDFAs. These high-power devices can be replaced in the future: DFB lasers can replace ECLs, with power consumption as low as 0.5 W. SOA can replace EDFA, with power consumption of only 0.8 W [2-4].

Figure.R18 Schematic diagram of the video transmission system power consumption.

The corresponding results and analysis are added to the revised manuscript and supplementary information:

(Supplementary Note IV) We also present the power consumption breakdown of the video transmission system as shown in Figure.S25. At the transmitter side, the switch served for signal routing consumes 43 W. The power consumption of two optical modules is 1.8 W each. The tunable ECL serves as the Tx LO with a power consumption of 50 W. The EDFA amplifies the mixed signals with a power consumption of 24 W. The amplified optical signal is then sent to the modified UTC-PD, consuming only 0.0044 W (2.2 V, 2 mA). At the receiver side, the signal is amplified by a low-noise amplifier (0.054 W) before sent into the TFLN modulator (0.045 W) for THz-to-optical conversion. The high-power laser driving the TFLN modulator is 20 W. The EDFA power consumption at the receiver side is 24 W. Two optical modules at the receiver side consumes 1.8 W each. And the power consumption of the Rx switch is 43 W. The overall system consumption is 211.3034 W. It is worth noting that in the current system, two of the major power consumption equipment are tunable ECL and EDFAs. These high-power devices can be replaced in the future: DFB lasers can replace ECLs, with power consumption as low as 0.5 W. SOA can replace EDFA, with power consumption of only 0.8 W [6-8].

Figure.S25 Schematic diagram of the video transmission system power consumption.

Reference:

- [1] Xie, Y. et al. Ultra-Wideband Tunable Thin-Film Lithium-Niobate-on-Insulator Microwave Photonic Filter. *ACS Photonics* 12, 1689–1697 (2025).
- [2] Fang, X. et al. Overcoming laser phase noise for low-cost coherent optical communication. *Nature Communications* 15(1), 6339 (2024).
- [3] Tao, Z. et al. Ultrabroadband on-chip photonics for full-spectrum wireless communications. *Nature*, 1-8 (2025).
- [4] Li, W. et al. 100-km polarization-orthogonal self-homodyne coherent WDM transmission using nonlinearity suppressed SOA. In *CLEO: Science and Innovations*, STu4G-5 (2023).

Comment 14: “Since many optical and RF amplifiers were used in this demo, please elaborate the system energy efficiency. Please provide a full power/energy budget (lasers, modulators/drive, UTC-PD bias, LNAs, DSP) and throughput/W figures for both fiber and wireless modes.”

Our reply: We agree with the reviewer that energy efficiency is an important metric for a communication system. Here, we analyze the system energy efficiency for both fiber and wireless modes. We present the power consumption breakdown of the fiber communication setup as shown in Figure.R19. The power consumption of the fixed wavelength laser is 1 W. An electrical power amplifier is used to amplify the signal generated from AWG consuming 2 W. The TFLN modulator consumes 0.045 W. The EDFA used to boost the modulated optical signal consumes 24 W. The power consumption of PD is 0.0012 W. When applying complex-biGRU algorithm, the DSP consumption is about 42.572 W. The overall system consumption is 69.5732 W, and the energy per bit is 69.6182 W/512 Gbps = 135.97 pJ/bit.

Figure.R19 Schematic diagram of the fiber communication system power consumption.

We also present the power consumption breakdown of the high-speed wireless communication setup as illustrated in Figure.R20. At the transmitter side, the fixed wavelength laser served as the signal carrier consumes 1 W. The power consumption of IQ modulator is 7.92 W, including drivers and automatic bias control. The tunable ECL serves as the Tx LO with a power consumption of 50 W. The EDFA amplifies the mixed signals with a power consumption of 24 W. The amplified optical signal is then sent to the modified UTC-PD, consuming only 0.0044 W (2.2 V, 2 mA). At the receiver side, the signal is amplified by a low-noise amplifier (0.054 W) before sent into the TFLN modulator (0.045 W) for THz-to-optical conversion. The high-power laser driving the TFLN modulator is 20 W. The EDFA power consumption at the receiver side is 24 W. Another tunable ECL acting as the Rx LO consuming 50 W. The power consumption of the complex-biGRU DSP algorithm for demodulating signals is 50.7 W. The overall link consumption is 227.7234 W. And the energy per bit is 227.7234 W/400 Gbps = 569.3 pJ/bit.

Figure.R20 Schematic diagram of the high-speed wireless communication system power consumption.

It is worth noting that in the current system, power consumption is primarily concentrated in tunable ECLs, EDFA and DSP. These high-power devices can be replaced in the future (Figure.R21): DFB lasers can replace ECLs, with power consumption as low as 0.5 W. SOA can replace EDFA, with power consumption of only 0.8 W [1-3]. Thus, overall system power consumption is projected to decrease to 62.3234 W, and the energy per bit will drop to 155.8 pJ/bit. Furthermore, high-speed data transmission without optical amplification by utilizing high-power DFB lasers has been demonstrated [4], revealing the potential for future EDFA/SOA-free transmission. The power consumption of the complex-biGRU algorithm can also be reduced through various optimizations as we mentioned in Comment 10.

Figure.R21 Schematic diagram of the future high-speed wireless communication system power consumption.

We have added the system energy efficiency analysis in the supplementary information:

(Supplementary Note III) We also analyze the energy efficiency of the fiber communication system. The power consumption breakdown of the system setup is presented in Figure.S14. The power consumption of the fixed wavelength laser is 1 W. An electrical power amplifier is used to amplify the signal generated from AWG consuming 2 W. The TFLN modulator consumes 0.045 W. The EDFA used to boost the modulated optical signal consumes 24 W. The power consumption of PD is 0.0012 W. When applying complex-biGRU algorithm, the DSP consumption is about 42.572 W. The overall system consumption is 69.6182 W, and the energy per bit is $69.6182 \text{ W}/512 \text{ Gbps} = 135.97 \text{ pJ/bit}$. Currently, power consumption is primarily concentrated in EDFA and DSP, which can be mitigated through various approaches. High-speed data transmission without optical amplification by utilizing high-power DFB lasers has been demonstrated [2], revealing the potential for future EDFA/SOA-free transmission. Power consumption of the complex-biGRU algorithm can also be further reduced through precision algorithm optimization and using ASIC chips [3].

Figure.S14 Schematic diagram of the fiber communication system power consumption.

(Supplementary Note IV) For energy efficiency analysis of the system, we present the power consumption breakdown of the high-speed wireless communication setup as illustrated in Figure.S22. At the transmitter side, the fixed wavelength laser served as the signal carrier consumes 1 W. The power consumption of IQ modulator is 7.92 W, including drivers and automatic bias control. The tunable ECL serves as the Tx LO with a power consumption of 50 W. The EDFA amplifies the mixed signals with a power consumption of 24 W. The amplified optical signal is then sent to the modified UTC-PD, consuming only 0.0044 W (2.2 V, 2 mA). At the receiver side, the signal is amplified by a low-noise amplifier (0.054 W) before sent into the TFLN modulator (0.045 W) for THz-to-optical conversion. The high-power laser driving the TFLN modulator is 20 W. The EDFA power consumption at the receiver side is 24 W. Another tunable ECL acting as the Rx LO consuming 50 W. The power consumption of the complex-biGRU DSP algorithm for demodulating signals is 50.7 W. The overall link consumption is 227.7234 W. And the energy per bit is $227.7234 \text{ W}/400 \text{ Gbps} = 569.3 \text{ pJ/bit}$.

Figure.S22 Schematic diagram of the high-speed wireless communication system power consumption.

It is worth noting that in the current system, power consumption is primarily concentrated in tunable ECLs, EDFA and DSP. These high-power devices can be replaced in the future (Figure.S23): DFB lasers can replace ECLs, with power consumption as low as 0.5 W. SOA can replace EDFA, with power consumption of only 0.8 W [6-8]. Thus, overall system power consumption is projected to decrease to 62.3234 W, and the energy per bit will drop to 155.8 pJ/bit. Furthermore, power consumption of the complex-biGRU algorithm can also be reduced through precision algorithm optimization and using ASIC chips [3].

Figure.S23 Schematic diagram of the future high-speed wireless communication system power consumption.

Reference:

- [1] Fang, X. et al. Overcoming laser phase noise for low-cost coherent optical communication. *Nature Communications* 15(1), 6339 (2024).
- [2] Tao, Z. et al. Ultrabroadband on-chip photonics for full-spectrum wireless communications. *Nature*, 1-8 (2025).
- [3] Li, W. et al. 100-km polarization-orthogonal self-homodyne coherent WDM transmission using nonlinearity suppressed SOA. In *CLEO: Science and Innovations*, STu4G-5 (2023).
- [4] Ostrovskis, A. et al. Optical Amplification-Free 400 Gbps Net Bitrate Links with a TFLN-based

Transmitter. In *Optical Fiber Communications Conference, M1G-1 (2025)*.

Comment 15: “What is the packaging scheme to integrate TFLN EO modulator and InP UTC-PD devices with fiber arrays and THz antennas while keeping the >200 GHz performance? In particular, please outline thermal management and connector constraints for ultra-high speed operation.”

Our reply: A practical THz packaging scheme for UTC-PD can follow the rectangular-waveguide output configuration commonly used in THz PD modules [1,2] to match with the band-pass THz antennas. The RF signal from the InP UTC-PD can be routed through short, parallel gold bonds into a quartz RF circuit that incorporates grounded-CPW transmission lines and an integrated bias-tee, and then coupled through a CPW-to-WR transition for efficient THz radiation.

For the DC to 200 GHz broadband scenario emphasized in this work, instead of the rectangular-waveguide, the same quartz interposer can connect to a 0.6 mm coaxial port [3], which is now commercially available and supports signal transmission up to 220 GHz. The RF transmission follows the same path as mentioned above, with a 0.6 mm end launch connector coupled to the coplanar waveguide (CPW). Since the UTC-PDs usually operate at relatively low currents and moderate powers, a ceramic submount can provide sufficient heat dissipation to maintain device temperature stability. Besides, the operational window of our modified UTC-PD can be extended up to 100 °C with minor temperature-induced variations as we discussed in Comment 5, demonstrating robust high-speed and linear operation under cruel environment.

For TFLN modulator packaging, a flip-chip bonding process with Au bumps could be utilized to integrate the TFLN modulator onto substrate with moderate dielectric constant (e.g., AlN, diamond), ensuring impedance continuity and minimal parasitic effects for >200 GHz operation. A low-loss substrate-integrated waveguide (SIW) or coplanar waveguide (CPW) is often used to deliver the high-frequency electrical signals to the modulator’s electrodes, avoiding wire bonds and maintaining signal integrity. Glass-insulated through-substrate vias (TSVs) can also be implemented to provide low-loss vertical electrical feedthroughs with minimal parasitic capacitance. To incorporate with the THz antenna, a smooth E-plane probe-based waveguide transition can be designed for the THz output to enable efficient coupling from the planar circuit to the waveguide, which has been widely adopted for UTC-PD packaging. To maintain the broadband performance from DC to 200 GHz, 0.6 mm coaxial connectors can be utilized with an optimized grounded CPW to microstrip transition, ensuring minimal reflection and modal resonance up to 200 GHz and beyond.

To address thermal challenges in packaging TFLN modulators, high-thermal-conductivity substrates (e.g., AlN, diamond) are usually utilized, enhanced with embedded thermal vias and optional micro-channel cooling for efficient heat dissipation. A low-stress, coefficient of thermal expansion (CTE) matched underfill material can also be used during flip-chip bonding to mitigate thermomechanical stress, while high-performance thermal interface materials (TIMs) can be applied between the chip and heat spreader to minimize thermal resistance and prevent performance degradation under high temperature conditions.

Reference:

- [1] Furuta, T. et al. D-band rectangular-waveguide-output uni-travelling-carrier photodiode module. *Electronics Letters*, 41(12), 715-716 (2005).
- [2] Ito, H. et al. Photonic millimetre-and sub-millimetre-wave generation using J-band rectangular-

waveguide-output uni-travelling-carrier photodiode module. *Electronics Letters*, 42(24), 1424-1425 (2006).

- [3] Martens, J. et al. Construction and initial studies on a 0.6 mm coaxial calibration kit to 220 GHz. In *2024 103rd ARFTG Microwave Measurement Conference (ARFTG)*, 1-5 (IEEE, 2024).

Comment 16: “The architecture uses all-photonic up/down-conversion with mirrored Tx/Rx OEO-derived LOs. Please quantify LO frequency-set accuracy, zero-IF alignment error, and capture range under temperature drift.”

Our reply: We thank the reviewer for pointing out concerns about the stability of the heterodyne beating, which was also raised by other reviewers. **We would like to clarify that our architecture does not employ OEO scheme. Instead, both the Tx and Rx sides utilize free-running tunable lasers as LOs.** Due to the influence of spontaneous emission and temperature drift, the frequency of the laser is not absolutely stable and exhibits frequency jitter. To quantify the Tx LO frequency-set accuracy, we characterize the free-running ECLs used in our experiment at Tx sides. As illustrated in Figure.R22a, the beat signal generated by two ECLs is captured by a commercial PD and analyzed using an ESA. We measure the frequency stability of the beat signal within 30 minutes and the beat signal drifts about 21 MHz as shown in Figure.R22b.

Figure.R22 (a) Schematic diagram and (b) 30 minutes results of the Tx LO frequency-set accuracy experiment.

For Rx zero-IF alignment stability, we employ the experimental setup in Figure.R23a for characterization. The beat signal generated by two Tx ECLs is loaded to an off-the-shelf modulator driven by another ECL laser. The Rx LO beats with the modulated sideband via a second PD, and the zero-IF alignment error is analyzed through the ESA. We test the frequency drift within 30 minutes and the maximum drift range is about 43 MHz, as shown in Figure.R23b.

Figure.R23 (a) Schematic diagram and (b) 30 minutes results of the Rx zero-IF alignment error experiment.

Although there is slight frequency drift between the free-running lasers, it does not affect the performance of the high-speed wireless communication system due to the wide capture range. We detune the Rx LO from zero-IF and demodulate the 70 Gbaud QPSK signal. The BER deteriorates with the frequency offset increasing as illustrated in Figure.R24. It can be seen that the system maintains acceptable performance with a capture range up to 8 GHz, far exceeding the 43 MHz frequency drift of

the lasers. From the results one can infer, for the ultrabroadband signal transmission, tens of MHz frequency drift of the lasers have negligible impact.

Figure.R24 Capture range of the proposed high-speed wireless communication system.

We have added the descriptions about frequency stability of our high-speed wireless communication system in the supplementary information:

(Supplementary Note IV) For the frequency stability of our system, we analyze the Tx LO frequency-set accuracy, Rx zero-IF alignment error and capture range. Due to the influence of spontaneous emission and temperature drift, the frequency of the laser is not absolutely stable and exhibits frequency jitter. To quantify the Tx LO frequency-set accuracy, we characterize the free-running ECLs used in our experiment at Tx sides. As illustrated in Figure.S19a, the beat signal generated by two ECLs is captured by a commercial PD and analyzed using an ESA. We measure the frequency stability of the beat signal within 30 minutes and the beat signal drifts about 21 MHz as shown in Figure.S19b.

Figure.S19 (a) Schematic diagram and (b) 30 minutes results of the Tx LO frequency-set accuracy experiment.

For Rx zero-IF alignment error, we employ the experimental setup in Figure.S20a for characterization. The beat signal generated by two Tx ECLs is loaded to an off-the-shelf modulator. The Rx LO beats with the modulated sideband via a second PD, and the zero-IF alignment error is analyzed through the ESA. We test the frequency drift within 30 minutes and the maximum drift range is about 43 MHz, as shown in Figure.S20b.

Figure.S20 (a) Schematic diagram and (b) 30 minutes results of the Rx zero-IF alignment error experiment.

Although there is slight frequency drift between the free-running lasers, it does not affect the

performance of the high-speed wireless communication system due to the wide capture range. We detune the Rx LO from zero-IF and demodulate the 70 Gbaud QPSK signal. The BER deteriorates with the frequency offset increasing as illustrated in Figure.S21. It can be seen that the system maintains acceptable performance with a capture range up to 8 GHz, far exceeding the 43 MHz frequency drift of the lasers. From the results one can infer, for the ultrabroadband signal transmission, tens of MHz frequency drift of the lasers have negligible impact.

Figure.S21 Capture range of the proposed high-speed wireless communication system.

To mitigate the impact of frequency drift on the system's performance, we can employ narrow-linewidth lasers with temperature control modules to enhance the stability of the beat signal. Various approaches have also been proposed to track and recover the frequency jitter in the transmission link, including self-coherent [4], time-domain pilots [5] and residual carrier modulation [6]. While these methods are demonstrated in fiber communication, they can be seamlessly transferred to wireless communication, simultaneously compensating frequency jitter of beat signals at both Tx and Rx ends.

Reference:

- [1] Shieh, W. et al. Carrier-assisted differential detection. *Light: Science & Applications*, 9(1), 18 (2020).
- [2] Olsson, S. L. et al. Record-high 17.3-bit/s/Hz spectral efficiency transmission over 50 km using probabilistically shaped PDM 4096-QAM. In *Optical Fiber Communication Conference*, Th4C-5 (2018).
- [3] Fang, X. et al. Overcoming laser phase noise for low-cost coherent optical communication. *Nature Communications* 15(1), 6339 (2024).

Comment 17: "We also anticipate more ...in monolithically fully integrated optoelectronic systems'. Please elaborate how to monolithically integrate lasers and other components on the TFLN platform?"

Our reply: Thanks for the comments. Recent years, with the continuous advancement of research in the TFLN field, not only electro-optic modulators but an increasing number of optoelectronic system components have been integrated onto the TFLN platform.

For light source generation, external-cavity-diode-laser with a III-V gain section and TFLN passive cavities has been proposed [1]. The external cavity is a Vernier mirror structure consisting of two racetrack resonators based on TFLN waveguides for laser mode selection. This design also enables fast on-chip reconfigurability, owing to TFLN's excellent electro-optic properties.

For electro-optic signal conversion, a photonic receiver consisting of a broadband antenna and a low-drive-voltage modulator monolithically integrated on TFLN has been reported [2]. The ultra-wideband antenna realizes a highly efficient direct millimeter wave to optical conversion from 27 GHz to 200 GHz. The on-chip antenna also relieves the demand for high-frequency electrical packaging, thereby enhancing the system integration level.

For signal manipulation, microwave photonics (MWP) based on TFLN platforms provides several advantages for RF signal processing, such as low power consumption, high signal fidelity, and large tuning range [3,4]. The programmable multifunctionality (e.g., band-notch filter, band-pass filter) of the TFLN MWP circuits also provides potential solutions to practical communication challenges, including channel selection, carrier filtering, and interference suppression.

For signal amplification, Erbium-doped lithium niobate on insulator waveguide amplifier (EDWA) with ultra-high internal net gain and large bandwidth electro-optic modulation has been proposed [5], unifying optical amplification and electro-optical dynamics at the same time.

For signal detection, ultra-wideband modified UTC PD integrated on TFLN with over 200 GHz OE bandwidth has been demonstrated [6], illustrating the feasibility for TFLN to serve as a complete platform for ultra-high-speed optical communication and photonic-enabled terahertz systems.

In summary, all the key components of high-speed optoelectronic systems have been individually integrated onto the TFLN platform. By combining these techniques, a monolithically fully integrated optoelectronic system for ultra-high-speed fiber-wireless communication can be expected.

We have added content about heterogeneous integrated photodetectors in the revised manuscript and rearrange the descriptions to follow the signal processing sequence.

(Main part, Page 8 line 43) We also anticipate more cooperations with integrated lasers [41], on-chip antennas [42], tunable microwave photonic filters [43,44], waveguide amplifiers [45], and heterogeneously integrated PDs [46] based on TFLN platform, eliminate discrete instruments, and culminate in monolithically fully integrated optoelectronic systems.

Reference:

- [1] Li, M. et al. Integrated Pockels laser. *Nature communications* 13, 5344 (2022).
- [2] Moller de Freitas, M. et al. Monolithically integrated ultra-wideband photonic receiver on thin film lithium niobate. *Communications Engineering* 4, 55 (2025)
- [3] Wei, C. et al. Programmable multifunctional integrated microwave photonic circuit on thin-film lithium niobate. *Nature communications* 16, 2281 (2025).
- [4] Xie, Y. et al. Ultra-Wideband Tunable Thin-Film Lithium-Niobate-on-Insulator Microwave Photonic Filter. *ACS Photonics* 12, 1689–1697 (2025).
- [5] Wang, Y. et al. A Microwave Photonic RF Receiver with Pre-amplification on Er-doped Lithium Niobate Platform. In *CLEO: Science and Innovations*, SS165_1 (2025).
- [6] Wang, L. et al. 230 GHz MUTC Photodiodes Integrated on Thin-film Lithium Niobate. In *Optical Fiber Communication Conference*, M2K-7 (2025).

Response to the report from the Referee #2

General comments: *“The manuscript titled ‘Integrated photonics enabling ultra-wideband fiber–wireless communication’ reports a seamless data path between optical fiber and wireless channels. The key components are thin-film lithium niobate modulators and uni-traveling-carrier photodetectors. The lithium niobate modulator achieves a record-high 3-dB bandwidth of 260 GHz, enabling a record-high single-channel data rate. When combined with a custom-designed unitraveling-carrier photodiode, the authors demonstrate record-fast fiber–wireless–fiber data communication.*”

Our reply: We sincerely appreciate the reviewer’s recognition of the performance of our work and are grateful for the reviewer’s constructive comments. Additional experiments and analysis have been conducted to elaborate the motivation and innovation of our work. The comments and our point-by-point responses are provided below.

Comment 1: “While the results advance the state of the art in terms of performance metrics, the contribution appears to be primarily an engineering refinement rather than a conceptual or scientific breakthrough. The modulator and photodiode designs largely follow established approaches [Ref. 27], and the wireless link demonstrated is limited to a 20-cm distance, which raises questions about its relevance to the envisioned applications. Moreover, it is unclear how the reported 8 mA photocurrent corresponds to 0 dBm of RF output power—clarification is needed on whether this represents the actual radiated power from the antenna.”

Our reply: We thank the reviewer for the comment. The key conceptual advance of this work lies in establishing a unified photonic communication architecture that bridges fiber and wireless domains, both at their state-of-the-art transmission speed, within a single integrated hardware platform. This approach delivers end-to-end ultrahigh-speed communication that can be deployed seamlessly across diverse environments, improving system’s environmental adaptability, scalability, and compactness. To enable this architecture, both the device and algorithm are specifically designed and optimized for hybrid fiber–wireless transmission. We make breakthroughs in both fundamental devices to broaden the bandwidth over 250 GHz and propose a new AI-based software algorithm with extraordinary signal equalization ability. As a proof-of-concept demonstration, record high transmission has been experimentally verified, including short-reach fiber interconnection, high-speed THz wireless transmission and multi-channel real-time videos transmission.

Furthermore, in this revised version, to further strengthen the practical applicability of our work, **we have conducted new experiments. Specifically, we achieve a 4-m distance 400 Gbps wireless high-speed transmission that shows no performance degradation compared with the previous 20-cm demonstration, confirming the robustness and extendibility of the system. In addition, by applying our TFLN modulator design methodology to 4-inch wafer-scale fabrication, we achieve what we believe to be the first wafer-scale production of TFLN modulators with bandwidths extending beyond 110 GHz up to 250 GHz, thereby showcasing the manufacturability and practical potential of our photonic platform.**

A detailed description of the innovations of each component in the system is presented below.

1) Ultra-wideband TFLN modulator:

(a) Ultra-wideband TFLN modulator with thorough impedance matching electrodes design.

To improve the EO bandwidth of the modulator, it is essential to minimize microwave losses while simultaneously achieving optimal velocity matching and impedance matching. In previous studies, RF loss and velocity matching of the electrodes have been thoroughly investigated. Many approaches such as quartz substrate and slow-wave electrodes have been proposed to address these two challenges. However, thorough impedance matching across all critical node of the device has often been overlooked, resulting in a degraded EO response, especially beyond 110 GHz. As shown in Figure.R25, the simulated results reveals that 25 Ω impedance mismatch will lead to 1 dB EO response degradation. Besides, impedance mismatch can also cause strong electrical reflection, leading to periodic resonance in the EO response. Moreover, pad areas between modulation electrodes and probes also impact the impedance matching, requiring careful design consideration. In this work, we extend the functionality of the slow-wave electrode from reducing microwave loss and achieving velocity matching to realize thorough impedance matching, simultaneously mitigating the RF losses while achieving almost perfect impedance match and velocity match across 1-220 GHz. The measured S11 and S21 parameters indicate the impedance matching performance of the electrodes, as we discussed in Reviewer #1, Comment 2.

Figure.R25 EO response degradation under different impedance mismatch.

(b) Wafer-scale demonstration of ultra-wideband TFLN modulator.

Beyond outstanding device performance, the manufacturing complexity and cost also need to be taken into consideration in practical applications. Although TFLN modulators with bandwidths exceeding 110GHz can also be realized based on photonic crystals or slow-light structures, they all utilize customized complex processes, significantly increasing manufacturing complexity and cost. In contrast, our solution complies with commercial wafer-scale fabrication and are capable for volume production. Building on the same wafer-scale quartz substrate TFLN platform with 110 GHz bandwidth developed by some of the authors [1], **we have now extended the bandwidth beyond 200 GHz using the above new design method through wafer-scale DUV lithography process. Compared with the EBL modulator, the DUV modulator shows no significant difference in EO response from 110 GHz to 220 GHz, confirming the massive scalability of our solution (Figure.R27).**

Figure.R26 4-inch wafer-scale quartz substrate TFLN platform.

Figure.R27 Comparison of 110 GHz - 220 GHz EO response between DUV and EBL process modulators.

We believe our scheme presents the first wafer-scale TFLN modulator with bandwidth exceeding 200 GHz, which are particularly suitable for both baseband fiber modulation and (sub)terahertz wireless modulation, as well as their seamless convergence. Such demonstration represents distinct contributions beyond previous work rather than an engineering refinement and is an indispensable component in all-optical high-speed fiber-wireless systems.

2) Ultra-wideband InP modified UTC-PD with high saturated power:

(a) RC-bandwidth enhancement through thick BCB dielectric layer.

In previous works, the parasitic capacitance between the coplanar signal pad and the buried n-contact (C_{con}) has been a major RC-limiting factor for high-speed PDs, as the air-bridge spacing cannot practically exceed $1 \mu\text{m}$. In this work, we introduced a thick $4 \mu\text{m}$ benzocyclobutene (BCB) dielectric layer beneath the CPW electrodes, specifically designed to minimize C_{con} and the capacitance between the CPW electrodes. This configuration effectively replaces the fragile air-bridge structure while providing both enhanced RC-limited bandwidth and improved mechanical robustness for devices with widths down to $2 \mu\text{m}$. As shown in Figure.R28, the measured total capacitance of devices with and without BCB is plotted as a function of mesa area. By linear fitting, the intercept corresponds to the parasitic capacitance (C_{st}) of each device type. The conventional air-bridge structure without BCB exhibits a parasitic capacitance of 20.3 fF , whereas introducing the thick BCB layer reduces it to 8.9 fF , confirming the effectiveness of this dielectric design in suppressing parasitic capacitance and minimizing the overall RC delay.

Figure.R28 Linear relationship between the total device capacitance and device area, where the intercept corresponds to parasitic capacitance.

(b) Enhanced saturation power through novel waveguide design.

Saturation power serves as a critical metric for wireless applications, as it determines the signal-to-noise ratio (SNR) and thus influences the achievable transmission distance and data rate. Waveguide-integrated photodiodes typically suffer from early saturation due to nonuniform optical absorption. To overcome this limitation, we innovatively extended the collector region outward to serve as the upper

cladding of the InGaAsP waveguide, thereby combining electrical design optimization with optical mode engineering. As shown in Figure.R29, the simulated optical field distributions reveal that, compared with the conventional structure without collector extension (Figure.R29a), the extended design (Figure.R29b) leads to smoother mode transition and more uniform optical absorption within the active region. This modification not only enlarges the waveguide mode size and reduces insertion loss, but also improves the coupling efficiency from the waveguide to the absorber and enhances the saturation output power. Consequently, the optimized UTC-PD achieves saturation performance comparable to, or even exceeding, that of conventional surface-illuminated photodiodes.

Figure.R29 Simulated optical field distributions of the UTC-PD with (a) conventional structure without collector extension and (b) proposed collector-extended structure.

Together, these design strategies yield a record-high and remarkably flat > 250 GHz bandwidth while maintaining >1 dBm saturation power and 0.24 A/W responsivity — a combination that has not been simultaneously achieved in any prior InP- or Si/Ge-based photodiode. The design strategies are specifically conceived to meet the stringent requirements of an ultra-wideband fiber–wireless system, representing a system-driven structural innovation rather than an incremental refinement.

For RF output power measurements, we use a heterodyne setup, where the generated RF signal arises from the optical beat between two lasers. The output power is collected by a GSG probe and measured by an RF power meter, rather than the radiated power from the antenna. Under ideal conditions, the optical modulation depth can reach 100 %, corresponding to complete conversion of the optical beat envelope into an RF current at the photodiode output. The purple solid line in Fig.2e represents this ideal case. The calculation assumes 100 % modulation depth and a 50 Ω load impedance using the following equation:

$$P_{\text{ideal}} = \frac{1}{2} R_L I_{ph}^2 \quad (R14)$$

According to this equation, a photocurrent of 8 mA yields an ideal RF output power of 2.04 dBm, consistent with the ideal line in Fig.2e. In our experiment, the UTC-PD exhibits -1 dB compression near 8 mA photocurrent, giving a measured maximum output power of 1.26 dBm at 140 GHz, which is about 1 dB below the ideal value.

3) Complex-biGRU algorithm:

Besides hardware breakthrough, AI-powered complex bidirectional gated recurrent unit (complex-

biGRU) algorithm has also been proposed. We expand the traditional GRU network into complex domain to adapt for both fiber and wireless scenarios. Besides, traditional NN-based equalization algorithm tends to squeeze the signals towards the boundaries of constellation diagrams, resulting in severe signal distortion, which is called 'jail window' effect (Figure.R30a and Figure.R30b). We implement a multi-level activation function before the output layer and 'jail window' effect is remarkably relieved (Figure.R30c). The proposed complex-biGRU algorithm shows extraordinary equalization capability even in the complicated channels and improves BER performance by several orders of magnitude. Detailed comparisons with other commonly used algorithms in terms of computational complexity, power efficiency, latency, and equalization capability are presented in Reviewer #1, Comment 7, 8 & 10.

Figure.R30 Constellation diagrams of 25 Gbaud 16-QAM based on the equalization using (a) DNN with multi-level activation function, (b) Complex-biGRU with ReLU activation function and (c) Complex-biGRU with multi-level activation function.

4) Ultra-wideband fiber–wireless communication system with longer range:

Facilitated by ultra-wideband and efficient E-O-E conversion, along with the advanced neural network algorithm, our system achieves high-quality data transmission across all-scenario telecommunication networks. State-of-the-art single-lane data rates of 512 Gbps with complex-biGRU and over 200 Gbaud DSP-free transmission are realized for short-reach fiber communication. For all-optical wireless links, record-high 400 Gbps THz transmission and 86 channels real-time 8K videos transmission have been achieved. While the demonstration in our original manuscript was performed at a 20-cm distance, several possible solutions to enhance transmission range have been mentioned in the Discussion section, such as high-gain antennas, PTFE lens and cascade THz amplifiers. **To meet the communication distance requirements of the envisioned applications, here we replace the origin horn antennas with the high-gain lens antenna and increase the transmission distance to over 4 m, with the setup shown in Figure.R31.** Figure.R32 depicts the transmission results of the 96 Gbaud QPSK and 76 Gbaud 16-QAM based on baseline DSP. After signal recovering, BER results meet the 20% SD-FEC threshold requirement. With the help of complex-biGRU algorithm, the transmission rate can be further increased to 100 Gbaud and 400 Gbps (Figure.R33). Distinguishable constellation diagrams of the 100 Gbaud QPSK and 100 Gbaud 16-QAM are observed and BER results maintains below the 20% SD-FEC threshold. In addition to high-speed THz communication, 8K real-time video transmission at a 4-meter distance has also been demonstrated, as shown in the Supplementary Video.

Figure.R31 Photograph of the 4-m experimental set-up.

Figure.R32 Constellation diagrams and BER results for (a) 96 Gbaud QPSK and (b) 76 Gbaud 16-QAM with baseline DSP method.

Figure.R33 Constellation diagrams and BER results for (a) 100 Gbaud QPSK and (b) 100 Gbaud 16-QAM with complex-biGRU algorithm.

Reference:

[1] Liu, Y. et al. Volume manufacturing of thin-film lithium niobate modulators with bandwidth > 110 GHz based on 4-inch wafer with a quartz handle. *Light: Advanced Manufacturing* 6, 17 (2025).

We hope the above statements can fully address the reviewer’s concern about the novelty of our work. To further strengthen the motivation and significance of our work, we have revised the manuscript and added the new experiments results.

(Supplementary Note I) Besides the extraordinary performance, our design also complies with commercial wafer-scale fabrication and are capable for volume production. We have fabricated wafer-scale TFLN modulators based on DUV lithography process using the same design parameters. Compared with the EBL modulator, the DUV modulator shows no significant difference in EO response from 110 GHz to 220 GHz, confirming the massive scalability of our solution (Figure.S6).

Figure.S6 Comparison of 110 GHz - 220 GHz EO response between DUV and EBL process modulators.

(Methods, Design and fabrication of integrated modulator and MUTC-PD) ...improving high-power performance. To further suppress parasitic capacitance and enhance RC-limited bandwidth, a 4 μm -thick BCB dielectric layer is introduced beneath the CPW electrodes, improving electrical isolation and device robustness. Moreover, the InP drift layer is extended outward to serve as the upper cladding of the InGaAsP waveguide, ensuring uniform optical absorption and mitigating localized saturation under strong light injection.

(Methods, Characterization of the EO/OE response of TFLN modulator and modified UTC-PD) For RF measurements, the RF signal generated by the UTC-PD is directly measured by a power meter through a GSG probe. Under 100 % modulation depth, the ideal RF output power follows Eq.6 with $R_L = 50 \Omega$, corresponding to the purple reference line in Fig.2e. Different measurement setups are used for different frequency ranges to ensure accurate power measurement.

$$P_{\text{ideal}} = \frac{1}{2} R_L I_{ph}^2 \quad (6)$$

(Main part, Page 6 line 21) We also verify the system's performance at a 4-m wireless distance using lens antennas. All BER results meet the SD-FEC threshold requirement with the highest transmission rate up to 192 Gbps for QPSK and 304 Gbps for 16-QAM respectively (Methods and Supplementary Note 4).

(Main part, Page 6 line 50) We also achieve 400 Gbps transmission for both QPSK and 16-QAM at a 4-m wireless distance using lens antennas (Methods and Supplementary Note 4).

(Main part, Page 6 line 105) We also conduct real-time video transmission at a 4-m wireless distance and clear live video can be seen on the receiver screen (Supplementary Video).

(Supplementary Note IV) To meet the communication distance requirements of the envisioned applications, we also verify the system's performance with the transmission distance increasing to 4 m. We replace the origin horn antennas with high-gain lens antennas. Figure.S26 depicts the transmission results of the 96 Gbaud QPSK and 76 Gbaud 16-QAM based on baseline DSP. After signal recovering, BER results meet the 20% SD-FEC threshold requirement. With the help of complex-biGRU algorithm, the transmission rate can be further increased to 100 Gbaud and 400 Gbps (Figure.S27). Distinguishable constellation diagrams of the 100 Gbaud QPSK and 100 Gbaud 16-QAM are observed and BER results maintains below the 20% SD-FEC threshold. In addition to high-speed THz communication, 8K real-time video transmission at a 4-meter distance has also been demonstrated, as shown in the Supplementary Video.

Figure.S26 Constellation diagrams and BER results for (a) 96 Gbaud QPSK and (b) 76 Gbaud 16-QAM with baseline DSP method.

Figure.S27 Constellation diagrams and BER results for (a) 100 Gbaud QPSK and (b) 100 Gbaud 16-QAM with complex-biGRU algorithm.

(Methods, Details of data transmission experiments) ...using different DSP techniques. For 4-m wireless transmission, we replace the origin horn antennas with 40 dBi gain lens antennas while keeping the rest of the link unchanged.

(Methods, Comparison of the UWB integrated photonics system)

Table IV. Comparison of high-speed THz wireless communication.

Method	Carrier frequency (GHz)	Carrier utilization efficiency (bit/s/Hz)	Distance (m)	Single channel data rate		Single channel symbol rate		Multi-channel capability
				Format	Rate (Gbps)	Format	Rate (Gbaud)	
All-optical	180	2.222	0.2 ^a	32-QAM	400	16-QAM	100	86
		1.333		16-QAM	240	QPSK	90	
		2.222	4 ^b	16-QAM	400	16-QAM	100	
		1.689		16-QAM	304	QPSK	96	
	101	0.707	20	64-QAM-OFDM	71.4	32-QAM-OFDM	11.9	1
	231	0.103	5	10-Nyquist-FDM	24	10-Nyquist-FDM	7.5	10
	288.5	0.173	16	QPSK	50	QPSK	25	1
All-electric	220	0.382	1260	16-QAM-OFDM	84	16-QAM-OFDM	21	4
	237.5	0.421	20	16-QAM	100	16-QAM	25	3
	465	0.563	10	PS-64-QAM	262	PS-64-QAM	46	1
Hybrid optoelectronic	100	0.640	1	16-QAM	64	16-QAM	16	1
	250	0.200	0.1	16-QAM	50	16-QAM	12.5	1
	340	0.353	3	8-QAM	120	8-QAM	40	1

^aConducted with horn antennas.^bConducted with lens antennas.

Comment 2: “Although the authors measured an impressive 3-dB bandwidth of 260 GHz, the optical–wireless link ultimately operates at ~100 Gbaud. This regime has been extensively explored in prior works across multiple platforms (e.g., Schuh et al., OFC Th5B.5; Lin et al., Opt. Express 27, 5610 (2019); Xu et al., Nat. Commun. 11, 3911 (2020)). While those studies primarily focused on fiber communication, the present manuscript does not convincingly demonstrate that the wireless integration offers a significant practical advantage. The applicability of the wireless demonstration to real-world scenarios remains uncertain.”

Our reply: We thank the reviewer for pointing out this concern. **In this work, we emphasize the first verification of seamless convergence of fiber-wireless architecture rather than focusing solely on ultra-wideband systems for either fiber or wireless communication.** For a 100 Gbaud optical–wireless link, it requires at least 50 GHz bandwidth in fiber and 100 GHz for wireless, totaling >200 GHz considering the guard band. Such large E-O-E bandwidth has not been achieved in previous studies. Even considering only fiber links or wireless links, we still achieve record-high single-channel symbol rates and data rates. As mentioned above, the wireless transmission distance has been experimentally extended to over 4 m, demonstrating progress toward real-world applications. **We believe our integrated photonic scheme will provide an effective solution to the applications that simultaneously require ultra-high-speed data transmission for both fiber and wireless scenarios, marking a critical step toward full-link high-speed latency-free all-optical telecommunication.** Specifically, we envision two practical deployment scenarios where our system holds strong potential for implementation:

1) 6G base station: The vision of the 6th Generation (6G) mobile communications will achieve a peak rate of up to Tbps and a high connection density supporting 10^7 terminal devices per square kilometer [1]. This requires base station deployment towards high-speed and ultra-dense development, which imposes tremendous pressure on centralized radio access network (C-RAN), especially on the fronthaul and backhaul links (collectively referred to as X-Haul) [2]. Optical fiber communication has become an ideal technology for X-Haul due to its large capacity and long transmission distance. However, the ‘all-fiber’ approach which laying fiber to every densely deployed base station poses significant challenges in terms of cost and deployment difficulty [3]. Furthermore, fixed fiber networks are constrained by limited topological flexibility, making it difficult to further meet the development needs of future communications. To overcome the limitations of all-fiber network, the fiber-wireless convergence architecture has emerged as a crucial pathway [3-5]. With this architecture, fiber and wireless communications leverage their synergistic advantages: the fiber network forms a robust, high-capacity backbone providing reliable long-distance transmission, while wireless technology addresses the “last mile” access challenge, offering dense deployment and flexible access capabilities. Although prior works have achieved transmission above 100 Gbaud in fiber and 50 Gbaud in wireless connection separately, they still cannot realize large-bandwidth fiber-wireless convergence transmission. Our scheme, however, fundamentally overcomes the device bandwidth bottleneck problem, enables the full utilization of THz-band spectrum resources and for the first time, achieves record-high transmission speed for both fiber and wireless scenarios. The high-speed and seamless fiber-wireless convergence network we realized will effectively empower the future C-RAN.

2) Wireless data center: With the development of the information society, particularly the exponential growth of artificial intelligence (AI), data centers have become an integral infrastructure in many enterprises [6,7]. Most data center networks (DCNs) today can be classified as wired networks, as copper cables and optical fibers are used for intra- and inter-rack connections. Despite great success,

wired DCNs face two inevitable challenges: deploying complexity and transmission rate. Many efforts have been made to improve the throughput of the data centers with the connection speed up to 400 Gbps [8,9]. However, these methods rely on fiber transmission, which significantly increases the deployment complexity and maintenance difficulty, particularly in future large-scale data centers. On the other hand, wireless data centers have been proposed in recent years to enhance the flexibility [10-14]. But current research only achieves transmission rates ranging from a few Gbps to several tens of Gbps, falling short of meeting the future interconnection demands of hundreds of Gbps or even Tbps. Our scheme synergistically integrates the dual advantages of optical communication's high bandwidth and wireless communication's flexibility. For chiplet and intra-rack interconnections, high-speed optical fiber connections could be deployed. While for inter-rack communication, optical assisted wireless links may be suitable to enhance interconnection flexibility while ensuring high-speed transmission through the utilization of THz frequency bands. The proposed ultra-wideband fiber-wireless system addresses the challenges of cabling complexity and transmission rate at the same time and will be a promising solution for future large-scale and high-throughput data centers.

We hope that the above-mentioned answer can provide the applicability to our solutions and address the reviewer's concerns. We have also added the following descriptions in the revised manuscript to clarify the applicability of our ultra-wideband fiber-wireless demonstration to real-world scenarios.

(Main part, Page 7 line 11) In this work, we propose and demonstrate a prototype system for UWB all-optical telecommunications, with record lane speeds demonstrated in both short-reach fiber and all-optical wireless links. **Such scheme will provide an effective solution to the applications that simultaneously require ultra-high-speed data transmission for both fiber and wireless scenarios, such as 6G base stations [35] and wireless data centers [36].**

Reference:

- [1] Tataria, H. et al. 6G wireless systems: Vision, requirements, challenges, insights, and opportunities. *Proceedings of the IEEE* 109(7), 1166-1199 (2021).
- [2] Zhang, Z. et al. 6G wireless networks: Vision, requirements, architecture, and key technologies. *IEEE vehicular technology magazine* 14(3), 28-41 (2019).
- [3] Townend, D. et al. Challenges and opportunities in wireless fronthaul. *IEEE Access* 11, 106607-106619 (2023).
- [4] Dat, P. T. et al. Seamless convergence of fiber and wireless systems for 5G and beyond networks. *Journal of Lightwave Technology* 37(2), 592-605 (2018).
- [5] Filgueiras, H. R. D. et al. Wireless and optical convergent access technologies toward 6G. *IEEE Access* 11, 9232-9259 (2023).
- [6] Minkenberg, C. et al. Co-packaged datacenter optics: Opportunities and challenges. *IET optoelectronics* 15(2), 77-91 (2021).
- [7] Terzi, C. et al. 60 GHz wireless data center networks: A survey. *Computer Networks* 185, 107730 (2021).
- [8] Wang, C. et al. Ultrabroadband thin-film lithium tantalate modulator for high-speed communications. *Optica* 11, 1614-1620 (2024).
- [9] Yamaguchi, Y. et al. Fully Packaged 100-GHz-Bandwidth EO-Equalizer-Integrated TFLN Modulator with Record-High Slope Efficiency Enabling 200-GBaud Signaling. In *Optical Fiber Communication Conference*, Th4D-4 (2025).
- [10] Cao, B. et al. Multiobjective 3-D topology optimization of next-generation wireless data center

network. *IEEE Transactions on Industrial Informatics* 16(5), 3597-3605 (2019).

- [11] Hamza, A. S. et al. Wireless communication in data centers: A survey. *IEEE communications surveys & tutorials* 18(3), 1572-1595 (2016).
- [12] Baccour, E. et al. A survey of wireless data center networks. In *2015 49th Annual conference on information sciences and systems*, 1-6 (2015).
- [13] Vardhan, H. et al. Wireless data center with millimeter wave network. In *2010 IEEE global telecommunications conference*, 1-6 (2010).
- [14] Rommel, S. et al. Data center connectivity by 6G wireless systems. In *2018 photonics in switching and computing*, 1-3 (2018).

Comment 3: “It is no doubt that the authors have a good device and got some really nice data. However, I do not think the level of innovation or the performance of the device warrants the paper’s publication in a high-impact journal like Nature.”

Our reply: We thank the reviewer’s recognition of our work and hope that the additional clarifications and data provided have addressed the concerns and substantiated the novelty of our works.

Response to the report from the Referee #3

General comments: “The authors present key integrated photonic components — specifically, an electro-optic (E/O) modulator and a photodetector — incorporated into a complete system that supports an impressive analog bandwidth exceeding 250 GHz. While previous works have demonstrated either plasmonic modulators or Ge-PIN photodetectors with even higher total electrical bandwidths (notably in Refs. [13] and [60]), those results were achieved using different technologies with inherent limitations. In contrast, the authors clearly outperform other related works in terms of both bandwidth and single-lane (channel) data rate for IMDD DSP-free, and all-THz communications. The practical applicability of the developed system is convincingly demonstrated by the successful transmission of 86 channels carrying 8K video streams across a spectral range from 138 GHz to 223 GHz.”

Our reply: We appreciate the reviewer’s recognition of the significance and performance of our work. We have provided a detailed response to clarify the technical aspects and address all raised concerns. We also conduct additional experiments to further support our claims.

Comment 1: “I have one question regarding the frequency stability of the resulting signal. Since a heterodyne up-conversion method was used both for the E/O frequency response measurement and for the up-conversion in the system demonstration, could the authors comment on the stability of the beat signal, considering that two independent lasers were used?”

Our reply: We thank the reviewer for pointing out concerns about the stability of the heterodyne beating, which was also raised by other reviewers. Due to the influence of spontaneous emission and temperature drift, the frequency of the laser is not absolutely stable and exhibits frequency jitter. To quantify the stability of Tx beat signal, we characterize the free-running ECLs used in our experiment at Tx sides. As illustrated in Figure.R34a, the beat signal generated by two ECLs is captured by a commercial PD and analyzed using an ESA. We measure the frequency stability of the beat signal within 30 minutes and the beat signal drifts about 21 MHz as shown in Figure.R34b.

Figure.R34 (a) Schematic diagram and (b) 30 minutes results of frequency stability test for Tx beat signal.

For Rx zero-IF alignment stability, we employ the experimental setup in Figure.R35a for characterization. The beat signal generated by two Tx ECLs is loaded to an off-the-shelf modulator driven by another ECL laser. The Rx LO beats with the modulated sideband via a second PD, and the zero-IF alignment error is analyzed through the ESA. We test the frequency drift within 30 minutes and the maximum drift range is about 43 MHz, as shown in Figure.R35b.

Figure.R35 (a) Schematic diagram and (b) 30 minutes results of frequency stability test for Rx beat signal.

Although there is slight frequency drift between the free-running lasers, it does not affect the performance of the high-speed wireless communication system due to the wide capture range. We detune the Rx LO from zero-IF and demodulate the 70 Gbaud QPSK signal. The BER deteriorates with the frequency offset increasing as illustrated in Figure.R36. It can be seen that the system maintains acceptable performance with a capture range up to 8 GHz, far exceeding the 43 MHz frequency drift of the lasers. From the results one can infer, for the ultrabroadband signal transmission, tens of MHz frequency drift of the lasers have negligible impact.

Figure.R36 Capture range of the proposed high-speed wireless communication system.

To mitigate the impact of frequency drift on the system's performance, we can employ narrow-linewidth lasers with temperature control modules to enhance the stability of the beat signal. Various approaches have also been proposed to track and recover the frequency jitter in the transmission link. In self-coherent scheme, an optical carrier co-propagates with the signal light through an additional dimension [1]. With accurate optical path matching between the carrier and signal, the laser phase noise can be effectively canceled. However, the self-coherent scheme typically requires a high carrier-to-signal power ratio (CSPR), thereby reducing the spectral efficiency of optical fibers and wasting EDFA gains. Another possible method is using time-domain pilots [2]. The pilot symbols periodically insert between the blocks of payload symbol to catch the phase rotation. Nevertheless, this scheme also reduces the spectral efficiency and cannot track rapid phase changes. Recently, residual carrier modulation (RCM) technique has been proposed to address the above challenges at the same time and enables the utilization of low-cost DFB lasers for conventional intradyne coherent transceiver [3].

While these methods are demonstrated in fiber communication, they can be seamlessly transferred to wireless communication, simultaneously compensating frequency jitter of beat signals at both Tx and Rx ends. We need to point out here that the above approaches have been proven effective in high-speed coherent optical wireless communication as demonstrated in an ongoing work by some of the authors in this work. The following figure shows the transmission results of 36 Gbaud QPSK signal using DFB laser sources with relatively large frequency drifting (Figure.R37). By employing the frequency jitter tracking technique, distinguishable constellation diagrams can be observed and BER remains at a constant level even without frequency offset estimation (FOE) and carrier phase estimation (CPE)

algorithms, thereby eliminating the influence of laser frequency shifts.

Figure.R37 Transmission results of 36 Gbaud QPSK with frequency jitter tracking scheme using DFB lasers.

We have added the descriptions about frequency stability of our high-speed wireless communication system in the supplementary information:

(Supplementary Note IV) For the frequency stability of our system, we analyze the Tx LO frequency-set accuracy, Rx zero-IF alignment error and capture range. Due to the influence of spontaneous emission and temperature drift, the frequency of the laser is not absolutely stable and exhibits frequency jitter. To quantify the Tx LO frequency-set accuracy, we characterize the free-running ECLs used in our experiment at Tx sides. As illustrated in Figure.S19a, the beat signal generated by two ECLs is captured by a commercial PD and analyzed using an ESA. We measure the frequency stability of the beat signal within 30 minutes and the beat signal drifts about 21 MHz as shown in Figure.S19b.

Figure.S19 (a) Schematic diagram and (b) 30 minutes results of the Tx LO frequency-set accuracy experiment.

For Rx zero-IF alignment error, we employ the experimental setup in Figure.S20a for characterization. The beat signal generated by two Tx ECLs is loaded to an off-the-shelf modulator. The Rx LO beats with the modulated sideband via a second PD, and the zero-IF alignment error is analyzed through the ESA. We test the frequency drift within 30 minutes and the maximum drift range is about 43 MHz, as shown in Figure.S20b.

Figure.S20 (a) Schematic diagram and (b) 30 minutes results of the Rx zero-IF alignment error experiment.

Although there is slight frequency drift between the free-running lasers, it does not affect the performance of the high-speed wireless communication system due to the wide capture range. We detune the Rx LO from zero-IF and demodulate the 70 Gbaud QPSK signal. The BER deteriorates with the frequency offset increasing as illustrated in Figure.S21. It can be seen that the system maintains acceptable performance with a capture range up to 8 GHz, far exceeding the 43 MHz frequency drift of the lasers. From the results one can infer, for the ultrabroadband signal transmission, tens of MHz frequency drift of the lasers have negligible impact.

Figure.S21 Capture range of the proposed high-speed wireless communication system.

To mitigate the impact of frequency drift on the system's performance, we can employ narrow-linewidth lasers with temperature control modules to enhance the stability of the beat signal. Various approaches have also been proposed to track and recover the frequency jitter in the transmission link, including self-coherent [4], time-domain pilots [5] and residual carrier modulation [6]. While these methods are demonstrated in fiber communication, they can be seamlessly transferred to wireless communication, simultaneously compensating frequency jitter of beat signals at both Tx and Rx ends.

Reference:

- [1] Shieh, W. et al. Carrier-assisted differential detection. *Light: Science & Applications*, 9(1), 18 (2020).
- [2] Olsson, S. L. et al. Record-high 17.3-bit/s/Hz spectral efficiency transmission over 50 km using probabilistically shaped PDM 4096-QAM. In *Optical Fiber Communication Conference*, Th4C-5 (2018).
- [3] Fang, X. et al. Overcoming laser phase noise for low-cost coherent optical communication. *Nature Communications* 15(1), 6339 (2024).

Comment 2: "Overall, the manuscript is well organized and meets all formal requirements. In my opinion, it makes a significant contribution to the advancement of converged optical/THz communication systems by demonstrating high-performance integrated components that hold promise for next-generation mobile networks, data centers, and beyond. Furthermore, the level of novelty is sufficient within the context of ultra-wideband microwave photonics systems. Therefore, I recommend the manuscript for publication."

Our reply: We sincerely thank again for the recognition of our work.

We appreciate the careful review by the reviewers and have modified the manuscript in accordance with their suggestions. Here, we present a point-by-point reply (in blue) to the reviewers' comments (in black), as well as the action taken (in red).

Response to the report from the Referee #2

General comments: "While I still think that the results shown in this work are mainly engineering refinement, I agree with the authors that the full-system demonstration is heroic and record-setting. In this sense, I agree that this work can be published in Nature."

Our reply: We appreciate the reviewer's recognition of our work and sincerely thank the reviewer for the thorough evaluation of our manuscript and constructive feedback.

Comment 1: "I just have one more minor question about the data. The EO and OE responses shown in Fig. 2 (c) and (f) are somewhat wiggly. I believe these are real features of the system rather than measurement artifacts. Are these features contributing to the raw error rate currently measured? If so, is the complex-biGRU algorithm effective at correcting for this error?"

Our reply: The fluctuating frequency response with frequency dips will cause more severe inter-symbol interference (ISI), leading to deterioration of the raw BER, particularly for higher-order modulation formats such as PAM-4 and 32-QAM. Our proposed complex-biGRU algorithm not only mitigates the spectral impairments induced by the modulator and PD, but also effectively compensates for linear and nonlinear distortions arising from other components in the transmission link. As shown in Fig.R1, the equalized BER improves by several orders of magnitude compared with the raw BER, which demonstrates the extraordinary equalization capability of the proposed complex-biGRU algorithm.

Figure.R1 Comparison of raw BER and BER after complex-biGRU for (a) NRZ, (b) PAM-4, (c) 16-QAM, and (d) 32-QAM.